# A quantitative approach to evaluating the GWP timescale through implicit discount rates

Marcus C. Sarofim[1], Michael R. Giordano[2]

[1]Climate Change Division, US Environmental Protection Agency, Washington, DC 20001, USA. orcid.org/0000-0001-7753-
1676
[2]AAAS S&T Policy Fellow Hosted by the EPA Office of Atmospheric Programs, Washington, DC 20001, USA.
orcid.org/0000-0002-6820-6668

*Correspondence to*: Marcus C. Sarofim (sarofim.marcus@epa.gov)

**Abstract.** The 100-year Global Warming Potential (GWP) is the primary metric used to compare the climate impacts of emissions of different greenhouse gases (GHGs). The GWP relies on radiative forcing rather than damages, assumes constant future concentrations, and integrates over a timescale of 100 years without discounting: these choices lead to a metric which is transparent and simple to calculate, but have also been criticized. In this paper, we take a quantitative approach to evaluating the choice of time-horizon, accounting for many of these complicating factors. By calculating an equivalent GWP timescale based on discounted damages resulting from $CH_4$ and $CO_2$ pulses, we show that a 100-year timescale is consistent with a discount rate of 3.3% (interquartile range of 2.7% to 4.1% in a sensitivity analysis). This range of discount rates is consistent with those often considered for climate impact analyses. With increasing discount rates equivalent timescales decrease. We recognize the limitations of evaluating metrics by relying only on climate impact equivalencies without consideration of the economic and political implications of metric implementation.

## 1. Introduction

The Global Warming Potential (GWP) is the primary metric used to assess the equivalency of emissions of different greenhouse gases (GHGs) for use in multi-gas policies and aggregate inventories. This primacy was established soon after its development in 1990 (Lashof and Ahuja, 1990, Rodhe 1990) due to its early use by the WMO (1992) and UNFCCC (1995). However, despite the GWP's long history of political acceptance, the GWP has also been a source of controversy and criticism (e.g., Wigley et al. 1998, Shine et al. 2005, Allen et al. 2016, and Edwards et al. 2016).

Key criticisms of the metric are wide ranging. Criticisms include the following: that radiative forcing as a measure of impact is not as relevant as temperature or damages (Shine et al. 2005); that the assumption of constant future GHG concentrations (Wuebbles et al. 1995, Reisinger et al. 2011) is unrealistic; that discounting is preferred to a constant time period of integration (Schmalensee, 1993); disagreements about the choice of time horizon in the absence of discounting (Ocko et al. 2017); that dynamic approaches would lead to a more optimal resource allocation over time (e.g., Manne & Richels 2001, Manne & Richels 2006), that the GWP does not account for non-climatic effects such as carbon fertilization or ozone produced by methane (Shindell 2015); and that pulses of emissions are less relevant than streams of emissions (Alvarez et al. 2012).

Unfortunately, including these complicating factors would make the metric less simple and transparent and would require reaching consensus regarding appropriate parameter values, model choices, and other methodology issues. The simplicity of the calculation of the GWP is one of the reasons that the use of the metric is so widespread.

In this paper, we focus on the choice of time horizon in the GWP as a key choice that can reflect decision-maker values, but where additional clarity regarding the implications of the time horizon could be useful. We also investigate the extent to which the choice of time horizon can incorporate many of the complexities of assessing impacts described in the previous paragraph. The 100-year time horizon of the GWP ($GWP_{100}$) is the time horizon most commonly used in many venues, for example in trading regimes such as under the Kyoto Protocol, perhaps in part because it was the middle value of the three time horizons (20, 100, and 500 years) analysed in the IPCC First Assessment Report. However, the 100-year time horizon has been described by some as arbitrary (Rodhe, 1990). The IPCC AR5 (Myhre et al. 2013) stated that "[t]here is no scientific argument for selecting 100 years compared with other choices". The WMO (1992) assessment has provided one of the few justifications for the 100-year time horizon, stating that "The GWPs evaluated over the 100-year period appear generally to provide a balanced representation of the various time horizons for climate response". Recently, some researchers and NGOs have been promoting more emphasis on shorter time horizons, such as 20 years, which would highlight the role of short-lived climate forcers such as $CH_4$ (Howarth et al. 2011, Edwards and Trancik, 2014, Ocko et al. 2017, Shindell et al. 2017). These studies each have different nuances regarding their recommendations – for example, Ocko et al. (2017) suggest pairing the GWP100 with the GWP20 to reflect both long-term and near-term climate impacts – and therefore there is no simple summary of the policy implications of this body of literature, but it is plausible that more consideration of short-term metrics would result in policy that weights near-term impacts more heavily. In contrast, some governments have suggested the use of the 100-year Global Temperature change Potential (GTP) based on the greater physical relevance of temperature in comparison to forcing, in effect downplaying the role of the same short-lived climate forcers (Chang-Ke et al. 2013, Brazil INDC, 2015). Therefore, the question of timescale remains unsettled and an area of active debate. We argue that more focus on quantitative justifications for timescales within the GWP structure would be of value, as differentiated from qualitative justifications such as a need for urgency to avoid tipping points as in Howarth et al. (2012).

While we argue that quantitative justifications for choosing appropriate GWP timescales are rare, as reflected by the judgment of the IPCC authors that no scientific arguments exist for selecting given timescales, there is a rich literature addressing many aspects of climate metrics. Deuber et al. (2013) presents a conceptual framework for evaluating climate metrics, laying out the different choices involved in choosing the measure of impact of radiative forcing, temperature, or damages, and temporal weighting functions that can be integrative (whether discounted or time-horizon based) or based on single future time points. Deuber et al. conclude that the Global Damage Potential (GDP) could be considered a "first-best benchmark metric", but recognize that the time-horizon based GWP has advantages based on limiting value-based judgments to a choice of time horizon, reducing scientific uncertainty by limiting the calculations of atmospheric effects to radiative forcing, and eliminating scenario uncertainty by assuming constant background concentrations. Mallapragada and Mignone (2017) present a similar framework and also note that metrics can consider a single pulse of a stream of pulses over multiple years. Several authors

have recognized that under certain simplifying assumptions, the GWP is equivalent to the integrated GTP, and therefore any timescale arguments that apply to analyses of one metric would also apply to the other (Shine et al. 2005, Sarofim 2012).

A few papers have applied GDP type approaches to evaluate the GWP in a manner similar to that of this paper. Boucher (2012) uses as uncertainty analysis similar to that used in this paper to estimate the GDP of methane. Boucher found that the GDP was highly sensitive to discount rate, over a range of 1 to 3%, and damage function over a range of polynomial exponents of 1.5 to 2.5, and that the median value of the GDP was very similar to the GWP100. Fuglestvedt et al. (2003) also used a GDP approach to map time horizons and damage function exponents to a discount rate, using IS92a as an emission scenario. Fuglestvedt et al. found that a discount rate of 1.75% and a damage exponent of 2 led to results equivalent to a GWP100. De Cara (2005), in an unpublished manuscript, also calculated the relationship between discount rates and time horizon, though they assumed linear damages.

An alternate approach is to evaluate metrics within the context of an integrated assessment model (IAM). There are several examples of such an approach. Van den Berg et al. (2015) analyse the implications of the use of 20 year, 100 year, and 500 year GWPs for $CH_4$ and $N_2O$ reductions over time within an IAM. The analysis estimated optimal costs to meet a 3.5 $Wm^{-2}$ target in 2100 and found that use of the $GWP_{20}$ and $GWP_{100}$ resulted in similar costs (within 4%) but that use of the $GWP_{500}$ resulted in higher costs by 18%. A key caveat here, as with many such analyses (including the present Sarofim and Giordano paper) is that the structure of the test can drive the evaluation result: in the case of van den Berg et al., the analysis ends in 2100, which will reduce the evaluated benefits of long-term metrics particularly for reductions that occur at the end of the century. These IAMs often use a discount rate of 5% for their net present value analysis. Other IAM analyses have concluded that changing the $CH_4$ to $CO_2$ ratio away from the $GWP_{100}$ has small effects on policy costs and climate outcomes (e.g., Smith et al. 2013, Reisinger et al. 2013). This is in large part because marginal abatement curves for $CH_4$ within these models have low-cost options (likely representing mitigation options such as landfill gas to energy projects and oil and gas leakage reduction) and high-cost options (reductions of enteric fermentation emissions from livestock) but few moderate-cost options. Therefore, for even a moderate carbon price, all the low-cost options will be enacted regardless of GWP, and no matter what the GWP, few high-cost options will be enacted. Such analyses may not fully consider non-market barriers or distributional effects for which changes in the GWP could be important.

While this paper focuses on a cost-benefit approach, there is also a potential need for cost-efficiency approaches, particularly in regard to stabilization targets such as 2 °C. However, a number of authors have argued that pulse-based metrics such as the GWP are not well-suited to achieve stabilization goals (Sarofim et al. 2005, Smith et al. 2012, Allen et al. 2016). Some actors (Brazil INDC, 2015) have claimed that certain metrics such as the Global Temperature Potential (Shine et al. 2005) or the Climate Tipping Potential (Jorgensen et al. 2014) are more compatible with a stabilization target such as 2 °C because they are temperature based. However, any pulse-based approach faces at least two major challenges related to stabilization scenarios. The first is that as a temperature target is approached, a dynamic approach will shift from favouring long-lived gas mitigation to favouring short-lived gases. While this shift may be optimal for meeting a target in a single year, it will be sub-optimal for any year after that year. The second challenge is that once stabilization has been achieved, any trading between

emission pulses of carbon dioxide and a shorter-lived gas will cause a deviation from stabilization. For example, trading a reduction in methane emissions for a pulse of $CO_2$ emissions will lead to a near term decrease in temperature, but also a long-term increase in temperature above the original stabilization level. One solution to the problem is a physically-based one. Allen et al. (2016) suggest trading an emission pulse of carbon dioxide against a sustained change in the emissions of a short-lived climate forcer. This resolves the issue of trading off what is effectively a permanent temperature change against a transient one. However, the challenge becomes one of implementation, as current policy structures are not designed for addressing indefinite sustained mitigation. A second solution is a dynamically updating global cost potential approach that optimizes shadow prices of different gases given a stabilization constraint (Tol et al. 2016), but again, implementation would be challenging. Alternatively, a number of researchers (Daniel et al. 2012, Jackson 2009, Smith et al. 2012) suggest addressing $CO_2$ mitigation separately from short-lived gases. Such a separation recognizes the value of the cumulative carbon concept in setting GHG mitigation policy (Zickfeld et al. 2009). However, this approach requires a central decisionmaker and loses the "what" flexibility that makes the use of metrics appealing (Bohringer et al. 2006). In economic terms, a temperature based target is equivalent to the assumption of infinite damage beyond that threshold temperature, and zero damages below that threshold (Tol et al. 2016).

This paper provides a needed quantification and analysis of the implications of different GWP time horizons. We follow the lead of economists who have proposed that the appropriate comparison between different options for GHG emissions policies is to compare the net present discounted marginal damages (Schmalensee 1993, Deuber et al. 2013). However, instead of proposing a switch to a GDP metric, we take the structure of the GWP as a given due to the simplicity of calculation and the widespread historic acceptance of its use. While other analysts have used similar approaches (Fuglestvedt et al. 2003, Boucher 2012), this paper reframes and clarifies key issues, and presents a framework for better understanding how different timescales can be reconciled with how the future is valued. The paper focuses on $CO_2$ and $CH_4$ as the two most important historical anthropogenic contributors to current warming, but the methodology is applicable to emissions of other gases and sensitivity analyses consider $N_2O$ and some fluorinated gases.

## 2. Methods

The general approach taken in this manuscript is to calculate the impact of a pulse of emissions of either $CO_2$ or $CH_4$ in the first year of simulation on a series of climatic variables. The first step is to calculate the perturbation of atmospheric concentrations over a baseline scenario. The concentration perturbation is transformed into a change in the global radiative forcing balance. The radiative forcing perturbation over time is used to calculate the impact on temperature and then damages due to that temperature change. Discount rates are then applied to these impacts to determine the net present value of the impacts. The details of these calculations are described here.

Concentrations: The perturbation due to a pulse of $CO_2$ is determined by use of IPCC AR5 equations (see Table 8.SM.10 from the IPCC AR5 assessment). The perturbation due to a pulse of $CH_4$ is calculated by the use of a 12.4 year

lifetime, consistent with Table 8.A.1 from IPCC AR5. In this manuscript, a pulse of 28.3 Mt of $CH_4$ is used (sufficient for a 10 ppb change in global $CH_4$ concentrations in the pulse year: results of a larger pulse are described in Sect. 3.3). The mass of the gas is converted to concentrations by assuming a molecular weight of air of 29 g/mole, and a mass of the atmosphere of $5.13 *10^{18}$ kg. These perturbations are added to baseline concentration pathways: for this study, we use the 4 RCP scenarios,

based on data from http://www.pik-potsdam.de/~mmalte/rcps/. This approach parallels the standard IPCC approach: however, various papers have noted that the lifetime of $CO_2$ presented in the IPCC includes climate carbon feedbacks, whereas the lifetime of $CH_4$ does not, which is a potential inconsistency (Gasser et al. 2017, Sterner and Johansson 2017). The discussion in Sect. 3.3 and 3.4 elaborates on the consequences of these choices.

       Radiative Forcing: The perturbation of radiative forcing from additional GHG concentrations are based on the

equations in Table 8.SM.1 from IPCC AR5. $CH_4$ forcing is adjusted by a factor of 1.65 to account for effects on tropospheric ozone and stratospheric water vapor, as is standard in GWP calculations. $N_2O$ forcing is adjusted by a factor of 0.928 to account for $N_2O$'s impacts on $CH_4$ concentrations, as is also standard in GWP calculations. Baseline radiative forcing is derived from the RCP scenario database.

       Temperature: Temperature calculations are all based on IPCC AR5 Table 8.SM.11.2. It should be noted that the

IPCC equations were designed for marginal emissions changes; therefore, using this approach to calculate temperatures resulting from the background RCP concentration pathways as well as the additional emissions pulses introduces a potential uncertainty. In order to calculate future temperatures, we also account for the present-day radiative forcing imbalance. Medhaug et al. (2017) suggest that this imbalance likely lies between 0.75 and 0.93 $W/m^2$. We use the mean (0.84 $W/m^2$) as the central estimate, and the range of this estimate in the sensitivity analysis presented above. The sum of the coefficients of

the equations in the IPCC temperature impulse response functions (1.06) is the sensitivity of the climate to an additional $W/m^2$: assuming that a doubling of $CO_2$ yields 3.7 W/m2, then the climate sensitivity implied by the IPCC suggested coefficients is 3.92. As a sensitivity analysis, the coefficients were scaled to yield climate sensitivities of 1.5 and 4.5 to mirror the likely range estimated by the IPCC.

       Damages: Damages as a percent of GDP were calculated by multiplying a constant by the square of the temperature

change since the baseline period. E.g., $D(2050) = a*\Delta T(2050)^2*GDP$. The net present value is then calculated using the discount rate such that $NPV(D(t)) = D(t)/(1+r)^{t-2010}$. Hsiang et al. (2017) present a recent justification for using a quadratic function for damages. For the sensitivity analysis, damage exponents of 1.5 or 3 were considered. Other formulations of the damage function have been considered in the literature. The first alternative is explicit calculation of damages within integrated assessment models. Another alternative is to include a higher power term in addition to the square exponent, so that at low

temperatures damages rise quadratically, but at high temperatures damages accelerate (Weitzman 2010). Finally, some analyses account for the impact of climate change on the economic growth rate, finding substantially higher damages (Dell et al. 2012, Moore and Diaz, 2015). The damage constant (a) (which cancels out in this particular application) as well as the GDP pathway are taken from the Nordhaus DICE model (Nordhaus 2017). Sensitivity analyses used a growth of 0.5 and 1.5 times that of the baseline growth for each five-year time period in the Nordhaus scenario. The GDP growth rates over the 21st century

from DICE (2.5%), and the high and low growth rate scenarios (1.3% and 3.8%), are consistent with the estimate of 21st century per-capita GDP growth from Christensen et al. (2018) of 2.1% (with a standard deviation of 1.1%), when added to the population growth rate of 0.4% from DICE (see the SI for a graph of the GDP scenarios). A temperature offset was also used because it is not clear what baseline temperature should be used for the damage function. A central value of 0.6 °C (the temperature change from 1951-1980 compared to 2011 based on the NASA GISStemp surface temperature record, GISTEMP 2016) is used, with sensitivities of 0 °C as a lower bound and 0.8 °C (the temperature change from 1880 to 2012 from the 2014 National Climate Assessment) as an upper bound. For the RCP3PD scenario, some future years (fewer than 1 out of 1000 of the total years considered across all sensitivities, and generally only for years near the end of the analysis) are cooler than the baseline temperature: in those years the net temperature change is set to zero to avoid numerical problems.

Discounting: Discount rates at 0.1% intervals between 0.5% and 15% were used in the analysis.

Equivalent GWP timescale: The above calculations produce a net present damages resulting from a pulse of $CH_4$ and for a pulse of the same mass $CO_2$. The ratio of these two values is a measure of the relative impact of $CH_4$ and $CO_2$. This measure of relative impact can be used to calculate the equivalent GWP timescale that would produce the same ratio.

## 3. Results
### 3.1. Evaluating the Climate Effects of an Emission Pulse of $CH_4$

The analysis starts by calculating the climate effects of an emission pulse of CH4. We introduce an emission pulse of 28.3 MT in 2011 (yielding a 10 ppb increase in CH4 concentration in the initial year) applied on top of the GHG concentrations of Representative Concentration Pathway (RCP) 6.0 (Myhre et al. 2013). Fig. 1 shows the changes in radiative forcing (RF; a), temperature (T; b), damages (c), and damages discounted at a 3% rate (d) out to the year 2300 resulting from such a pulse. Fig. 1 relies on calculations that use central estimates of the uncertain parameters, as discussed in the Methods section. While the graph is truncated at 2300, the calculations used in this paper extend to 2500. The impacts of an emission pulse of $CO_2$ are also shown, using 24.8 times the mass of the $CH_4$ pulse (this factor is chosen to create equivalent integrated damages over the full time period when discounted at 3% as shown in Fig. 1d). Fig. 1a and 1b demonstrate the tradeoffs between near-term and long-term impacts when assigning equivalency to emission pulses of different lifetimes. After 100 years, the radiative forcing effects of the $CH_4$ pulse decay to 0.04% of the initial forcing in the year of the emission pulse, and the temperature effects decay to 4% of the peak temperature (reached 10 years after the pulse). In contrast, after 100 years the radiative forcing effects of the $CO_2$ pulse decay to 22% of the initial forcing, and the temperature effects decay to 51% of the $CO_2$ peak temperature (reached 18 years after the pulse). The immediacy of the temperature effects for the $CH_4$ pulse creates larger damages in both overall and discounted dollar terms for the first 42 years. After 43 years, the sustained $CO_2$ effects overtake the $CH_4$ effects. With a different discount rate, a different factor would have been used to calculate the $CO_2$ mass used for the $CO_2$ pulse, which would change the crossing point for damages – a higher discount rate would require a larger $CO_2$ equivalent pulse relative to the $CH_4$ pulse, and therefore an earlier crossing point (and vice versa). Fig. 1c demonstrates the dramatic increase in damage

over time due to the relationship of damage to economic growth. In the case of $CH_4$, damage peaks in 2032 and declines until 2080 as a result of the short lifetime of the gas. The increase in damages after 2080 is due to the component of the temperature response function that includes a 409 year timescale decay rate, such that after 100 years the decrease in the $\Delta T^2$ component of the damage equation is about 0.5%/year, and because that decay rate is slower than the rate of GDP growth, net damages

grow. Fig. 1d demonstrates the dramatic decrease in future damages when applying a constant discount rate. Taken as a whole, these 4 figures demonstrate the tradeoffs required when attempting to create equivalences for emissions of gases with very different lifetimes.

### 3.2. Implying a Discount Rate

This analysis of evaluating the radiative forcing, temperature, damages and discounted damages of a pulse emission can be used to calculate the consistent GWP timescale for a given discount rate or, conversely, the discount rates that are consistent with a given GWP timescale by comparing the net present discounted marginal damages of $CH_4$ to $CO_2$. Figure 2 shows the relationship between the discount rate and the GWP timescale. Here we focus on what discount rates are consistent with a GWP time horizon in order to show the discount rates implied by common choices of GWP timescales. The converse

calculation is relevant for an audience that has a preferred discount rate and is interested in the implied GWP timescale.
From Fig. 2, the discount rate implied by the GWP 100 is 3.3% (interquartile range of 2.7% to 4.1%). The discount rate implied by a 20-year GWP timescale is 12.6% (interquartile range of 11.1% to 14.6%). Results in the figure are truncated to the year 2300 and the calculation is truncated to the year 2500, which may matter at very low discount rates due to the long lifetime of $CO_2$. At a 3% discount rate, 90% of the discounted $CO_2$ damages from an emissions pulse comes in the first 157 years, and

95% in 189 years. For $CH_4$, the equivalent 90/95% is 87/123 years, with the long tail on temperature effects causing elongated damages beyond the lifetime of the gas itself. Even at a 2% discount rate, 95% of the $CO_2$ damages come in the first 287 years. At discount rates lower than 2%, however, truncation effects can account for errors in damage ratio estimates of greater than a percent, indicating that longer calculation timeframes may be necessary to capture the full effect of the emissions pulse.
There is much discussion regarding which discount rate are most appropriate for use in evaluating climate damages. Since

2003, the US Government has used discount rates of 3% and 7% to evaluate regulatory actions, where 3% was deemed appropriate for regulation that "primarily and directly affects private consumption" and 7% for regulations that "alter the use of capital in the private sector" (OMB, 2003). From the current analysis, a 3% discount rate is consistent with a GWP of 118 years (interquartile range of 84-171 years) and 7% with a GWP of 38 years (interquartile range of 32-47 years). The OMB Circular also recognizes that there are special ethical considerations when impacts may accrue to future generations, and

climate change is a prime example of an impacts where discount rates lower than 3% could be justified. A number of researchers have advocated for time-dependent declining discount rates (Weitzman 2001, Newell and Pizer 2003, Gollier et al. 2008). The UK and France both already use declining discount rates in policymaking, and in both cases, the certainty equivalent discount rate drops below 3% within a hundred years, and approaches 2% within 300 years (Cropper et al. 2014).

This manuscript does not select a single "correct" discount rate. However, the analysis shows that the 100-year timescale is consistent, within the interquartile range, with the 3% discount rate that is commonly used for climate change analysis. In contrast, a 20-year time horizon for the GWP implies discount rates larger than those used in any climate change analysis publications to date.

### 3.3. Sensitivity Analyses

Figure 2 shows the median, interquartile, interdecile, and extremes of the equivalent GWP time horizon corresponding to a given discount rate from a sensitivity analysis. The uncertainty was calculated assuming equal likelihood of each of the 972 combinations of all of the parameter choices used in this manuscript: 4 RCPs, 3 climate sensitivities, 3 damage exponents, 3 forcing imbalance options, 3 temperature offsets, and 3 GDP growth rates. The ranges chosen for each parameter are described in the methods. The parameters with the largest effect on the uncertainty of the calculated GWP (at a discount rate of 3%) are the rate of GDP growth and the damage exponent (see Table 1). For these 6 parameters, the choices that lead to larger damages from $CH_4$ relative to $CO_2$ are a low GDP growth, a low damage exponent, a low emissions scenario, a higher temperature offset (e.g., assuming that damages are a function of warming from preindustrial, not warming from present day), a lower climate sensitivity, and a higher current forcing imbalance. The general trend is that the more that damages are expected to grow in the future (e.g., high GDP growth, damage exponent, or emissions scenario), the longer the equivalent timescale is for a given discount rate.

While $CO_2$ and $CH_4$ are the largest contributors to climate change (as evaluated by contributions of historical emissions to present-day radiative forcing as in Table 8.SM.6 in the IPCC, or by the magnitude of present-day emissions as evaluated by the standard $GWP_{100}$), it is also informative to evaluate emissions of other gases with these techniques. Table 2 shows 5 gases and their atmospheric lifetimes. For each gas, an "optimal" GWP timescale was calculated that would replicate the ratio of net present damage of that gas to $CO_2$ at a discount rate of 3%. The ratio of the $GWP_{100}$ and the $GWP_{20}$ to that optimal damage ratio is also shown. For longer-lived gases (e.g. $N_2O$ and HFC-23), there is no integration time period that can produce a ratio as large as the calculated damage ratio at a discount rate of 3%. For these gases, we list the timescale that yields the maximum possible ratio and note that the GWP for longer lived gases is fairly insensitive to timescale (further comparisons of non-$CO_2$ gases are presented in the SI). This table shows that at a discount rate of 3%, and as evaluated using net present damage ratios, the use of a 100-year timescale is consistent (interquartile range) with the optimal timescale/damage ratios for methane. For gases with lifetimes in centuries, the GWP at any timescale undervalues these gases, but the magnitude of that undervaluation is somewhat insensitive to the choice of timescale. For the longest lived gases, the GWP also undervalues reductions in these gases, but the longer the timescale the better the match.

In addition to investigating the sensitivity of these results to different choices of the six listed parameters, and five different gases, several other sensitivity experiments were performed. These experiments were chosen to investigate whether certain assumptions are important, as well as alternate approaches to constructing the model.

The first set of experiments involve analysis choices that end up having little difference in terms of timescale estimation. In general, this is because changes in these choices effect both the GWP and the damage estimation equally, and therefore cancel out. One experiment involved changing the size of the emissions pulse to 373 MMT (about one year's anthropogenic emissions according to Saunois et al. 2016). The effect on damage ratios of this change was less than 1%. Another experiment involved doubling the radiative efficiency of methane: while this led to a doubling of the estimated damage ratio, it also led to a doubling of the estimated GWP, such that the change in estimated timescale was about 1/10[th] of 1%. This experiment confirms that timescale estimates are insensitive to updates to estimates of the radiative efficiency of individual gases (such as the finding of Etminan et al. 2016 that methane has greater forcing effects than previously estimated). A third experiment arose because of the question of consistency between the treatment of $CO_2$ and $CH_4$ in terms of climate-carbon feedbacks (Gasser et al, 2017, Sterner and Johansson 2017). Using the $CO_2$ lifetime from Gasser et al. (2017) without climate-carbon feedbacks, an increase in damage ratios of about 8% was estimated, but a similar increase in GWP of about 7% was estimated, with a net effect on timescales of less than 1%. The converse experiment (including climate-carbon feedbacks in both the $CO_2$ and $CH_4$ lifetimes) was not analysed due to the increased complexity of the calculation. However, given that the virtue of the GWP is its simplicity, the authors suggest that the use of lifetimes without climate-carbon feedbacks for either gas should be preferred over the inclusion of those feedbacks in the lifetimes of both gases (Sarofim 2016).

Another experiment considered the use of a Ramsey type framework for discounting future damages. The use of such a framework has been recommended by the National Academies (NAS, 2017). In this framework, discount rates are a function of the marginal utility of consumption, the pure rate of time preference, and the future growth rate of per capita consumption. It is the latter dependence that makes this sensitivity analysis particularly interesting, as this pairs higher consumption growth (leading to higher damage ratios) with higher discount rates (leading to lower damage ratios). For this experiment, the Ramsey parameters were calibrated to yield an average discount rate for the reference GDP of 5% over the first 30 years of the analysis, given a pure rate of time preference of 0.01%. Under this assumption, the median timescale under the reference GDP scenario increases to 135 years, because even though the initial discount rates are higher than 3%, over the entire period of the analysis the average discount rate is only 1.5%. However, unlike in the original analysis, under the high GDP growth scenario the damage ratio increases and the equivalent timescale decreases to 90 years, because the increase in discount rate resulting from high growth has a larger effect on damages than the long-term increase in GDP (and vice versa for low GDP growth). The difference between the damage ratios for the high and low GDP growth scenarios is still about a factor of 2. A future analysis could pair GDP scenarios with emissions scenarios to take into account the potential correlation of the two.

Boucher (2012) and Fuglestvedt et al. (2003) both applied similar approaches to the one used in this paper, but both papers identified a discount rate consistent with the GWP100 that was somewhat lower than the median 3.3% value found in this paper. The most evident different between the approach in these previous papers and this article is that this article assumes that damages are expressed as a percent of GDP, and the previous analyses did not. In order to more closely emulate the Boucher and Fuglestvedt approach, the model was tested by using constant GDP over the entire time period, and the GWP100

was found to be most consistent with a discount rate of 1.2% (interquartile range of 1.0 to 1.9%) in contrast to 3.3% (interquartile range of 2.7% to 4.1%).

Myhre et al. (2013) justified exclusion of the 500 year GWP based on the large uncertainties and ambiguities involved with far future projections. This analysis extends through 2500, and therefore might be subject to some of those same uncertainties. Therefore, the effect of two shorter time periods were investigated. When truncating the analysis after 150 years, the GWP100 was still found to be consistent with a discount rate of 3.3%, with the upper interquartile bound also remaining constant at 4.1%, though the lower end of the interquartile range decreased modestly to 2.4%. When the analysis was truncated at 100 years into the future, the implicit discount rates dropped more substantially, to 2.6% (interquartile range of 1.5% to 3.5%). Truncating the analysis will naturally make $CH_4$ mitigation appear more favourable relative to $CO_2$, but even discount rates as small as 3% are sufficient to make effects more than 150 years into the future inconsequential to the results.

A final experiment considered the inclusion of damages due to rate of change as well as due to absolute temperature. Inclusion of rate of change damages has had important influences on previous analyses. For example, in Manne and Richels (2001), the dynamic optimization solution for approaching a temperature threshold placed little value on $CH_4$ reduction relative to $CO_2$ until a couple decades before the threshold was reached: but when a rate of change requirement was added, the relative value of $CH_4$ reduction stayed fairly constant over the time period. The challenge for this analysis is in determining the appropriate damage form, as the literature for estimating damages due to rate of change is not as robust as for absolute changes. As a test case, the peak rate of change damages under the median parameter values were calibrated to be equal to the absolute damages in the year 2060 (50 years into the analysis). The effect of inclusion of this effect was to increase the damage ratio of $CH_4$ to $CO_2$ by 2.4%. This fairly modest impact is consistent with results of Bowerman et al. (2013) or Rogelj et al. (2015) which suggest that near-term mitigation of SLCFs have modest effects on reducing the peak rate of change for higher future emissions scenarios, and that delayed SLCF mitigation may yield most of the same benefits as immediate SLCF mitigation in terms of both peak absolute change as well as rate of change. In order examine how this effect could be sensitive to a lower emissions scenario, the analysis was repeated for the RCP3PD scenario by itself. Under this assumption, the damage ratio increases by 53%, resulting in a decrease of the optimal timescale for RCP3PD associated with a discount rate of 3% from 94 years to 54 years. This result is also consistent with Bowerman et al. which found more benefit to reducing near-term SLCF emissions if future emissions are expected to be low.

### 3.4. Additional Uncertainties

There are a number of uncertainties involved in this analysis. They can be divided into three categories: those that may change the relative climate-related discounted damages of $CH_4$ compared to $CO_2$ but have minimal effect on the implied timescale of the GWP, those that have a large impact on the implied timescale, and those effects of $CH_4$ and $CO_2$ that are unrelated to their climate forcing.

As shown above, uncertainties in this analysis that do not have a large impact on the calculated GWP timescale include factors that have similar effects on the GWP and the $CH_4$:$CO_2$ discounted damage ratio, such as radiative efficiency or consistent treatment of climate-carbon feedbacks.

In contrast, the timescale of ocean heat uptake, the lag between the timing of atmospheric temperature response to forcing and the response of sea level (e.g., Zickfeld et al. 2017), and other issues that are inherent to the timing of climate impacts—but are not necessarily included in the GWP calculation—might all affect the implied timescale. One potential way to explore some of these effects would be to use a more complex climate model to evaluate the radiative forcing and temperature effects of the emission pulses. The shape of the damage function can also have a substantial effect: different exponents for the polynomial form were tested, as was the inclusion of rate of change, but the full range of possible damage functions is substantially larger, including multi-polynomial behavior (Weitzman, 2001) or the potential for persistent influences on economic growth (Burke et al. 2015).

An additional category of effects has less relevance to an analysis of appropriate timescale of climate impacts, but would be important for overall valuation. These are generally gas-specific effects which should most appropriately be considered on a case-by-case basis rather than folding into a timescale analysis that will influence the mitigation choices for all gases. One example is the inclusion of $CO_2$ fertilization effects, which would reduce the relative importance of decreasing $CO_2$ compared to other gases. Other examples include the health effects of $O_3$ produced by reaction of $CH_4$ in the atmosphere (Shindell et al. 2015, Sarofim et al. 2017), $CO_2$ effects on ocean acidification, or the possible reduced efficacy of $CH_4$ compared to $CO_2$ (Modak et al. 2018). These effects can be important for making mitigation decisions but are outside of the scope of consideration for a manuscript focusing on how to choose a time horizon for comparing climate impacts. As an example, if the solution to undervaluing $CH_4$ mitigation due to its $O_3$ effects is to reduce the appropriate timescale for GHG comparisons, an identical gas without $O_3$ chemistry implications would be similarly prioritized. One potential approach which could be explored might be to apply a multiplier to the GWP after calculation to take into account these non-climatic effects, much like the GWP of methane takes into account indirect effects on climate through the production of tropospheric $O_3$ and stratospheric $H_2O$ by the use of a multiplicative factor.

### 3.5. Caveats

The analysis presented here suggests that the use of a 100-year time horizon for the GWP is in good agreement with what many consider an appropriate discount rate; however, we offer several caveats. Most importantly, this analysis makes the assumption that the net present damage of $CH_4$ and $CO_2$ is the best metric for evaluating relative impact of gases. When analysing several different common metrics, Azar and Johansson (2012) asked whether society would prefer integrated metrics such as the GWP, single time period metrics such as the GTP, or economic metrics such as the global damage potential which is parallel to the metric given primary weight in this paper. Considering the applications of a metric within the context of an integrated assessment model could enable analysis of more complex economic interactions. Alternatively, a decision-making framework might consider factors other than damages: for example, in a multi-stage decision-making process under

uncertainty, it might be possible that long-lived gas mitigation should be prioritized in order to increase future option-value. Or there might be reasons to prioritize mitigation options that apply to capital stocks with long lifetimes or to decisions which involve path dependence, as those decisions would be more costly to reverse in the future.

This metric approach is also not designed to achieve a long term temperature goal such as stabilization at 2 °C above
preindustrial temperatures. We note no metric designed to tradeoff emission pulses is consistent with stabilization: one solution to this dilemma is the GWP* introduced by Allen et al. (2016) which creates an equivalence between an emission pulse of $CO_2$ and a constant stream of $CH_4$. This analysis only looks at a pulse of emissions in 2011, and does not examine whether the equivalent timescale might change over time.

## 4. Conclusions

This analysis uses a global damage potential approach to calculate the implicit discount rate corresponding with different GWP timescales. While this is not the first analysis to calculate the implicit discount rate of the 100 year GWP (Boucher 2012, Fuglestvedt et al. 2003), the framework presented here allows for a more complete and wide-ranging analysis of sensitivities than has been presented previously, and the connection between the timescale and the implicit discount rate is made more
clearly. The 100 year GWP is the inter-gas comparison metric with the widest use, and the results presented here show that the 100-year timescale is consistent with an implied discount rate of 3.3% (interquartile range of 2.7% to 4.1%). Alternatively, the 3% discount rate used for calculating social damages in some regulatory impact analysis contexts is consistent with timescales of 84-171 years. The uncertainty range in the results is most sensitive to the assumptions regarding future GDP growth and to the choice of exponent in the damage function. These results are insensitive to assumptions regarding radiative efficiency,
pulse size, or consistent treatment of climate-carbon feedbacks. At discount rates of 3% or higher, the analysis can be truncated to 150 years (rather than the default calculation through 2500) with little effect. Inclusion of damages resulting from the rate of change in addition to absolute temperature changes has little effect except in the case of a low emissions future, where it results in a decrease in the timescale consistent with a 3% discount rate to 54 years. Applying the methodology in this paper to calculate the implied intertemporal values of a 20 year GWP, a timescale that has received some recent attention, results in
an implicit discount rate of 12.6% (interquartile range of 11.1% to 14.6%).

These results provide support to the contention that 100 years is a reasonable timescale choice for the GWP, given the assumption that the relative climate damage of pulses of different greenhouse gases is an appropriate means of valuation, and that the 3% discount rate is a reasonable measure of the value of the future. This finding is robust to a number of sensitivity analyses. In contrast, the analysis suggests that the 20 year GWP timescale is most consistent with an implicit discount rate
much higher than the standard social discount rate, except in scenarios with low future emissions and high rate of change damages, similar to concerns expressed in other analyses (Shoemaker and Schrag, 2013). However, while the implicit timescale was derived from analysing the climate impacts resulting from $CH_4$ emissions relative to $CO_2$ climate impacts, the results do not necessarily inform a specific relative importance of $CH_4$ mitigation compared to $CO_2$. Such a relative importance calculation should take into account the latest research on radiative efficiencies (Etminan et al. 2016). and could potentially

also take into account non-climate impacts like the health effects of $CH_4$-derived $O_3$ (Shindell, 2015, Sarofim et al. 2017). Inclusion of non-climate impacts could perhaps use an adjustment factor in the same way that the $CH_4$ GWP already includes adjustment factors for the climate effects of $CH_4$-derived $O_3$. Additionally, the appropriate GWP timescales can also be informed by the manner in which the metric is being used for policy or informational purposes.

The methodology presented here is transparent (the code is available in the SI), rigorous (the parameters and functional forms are derived from respected sources), and flexible (as demonstrated by a wide range of sensitivity analyses from inclusion of rate-of-change damages to Ramsey discounting). This framework can be a valuable resource for quantitatively examining appropriate timescales given different assumptions about discounting, the relationship of damages to both absolute and rate of temperature changes, tipping points, future emissions scenarios, and other factors.

**Code Availability**

The R code used in developing this manuscript has been submitted as supplemental information.

**Author Contributions**

Both MCS and MRG contributed to experiment design, coding, figure development, and manuscript writing.

**Competing interests**

The authors declare that they have no conflict of interest.

**Disclaimer**

This publication was developed under Assistance Agreement No. X3-83588701 awarded by the U.S. Environmental Protection Agency to AAAS. It has not been formally reviewed by EPA. The views expressed in this document are solely those of the authors and do not necessarily reflect those of the Agency. EPA does not endorse any products or commercial services mentioned in this publication.


## Acknowledgements

M.R.G. was supported as a Science and Technology Policy Fellow by the American Association for the Advancement of Science (AAAS) STPF program. The authors would like to thank numerous colleagues at the EPA for their thoughts and discussions regarding GHG metrics and climate economics.

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

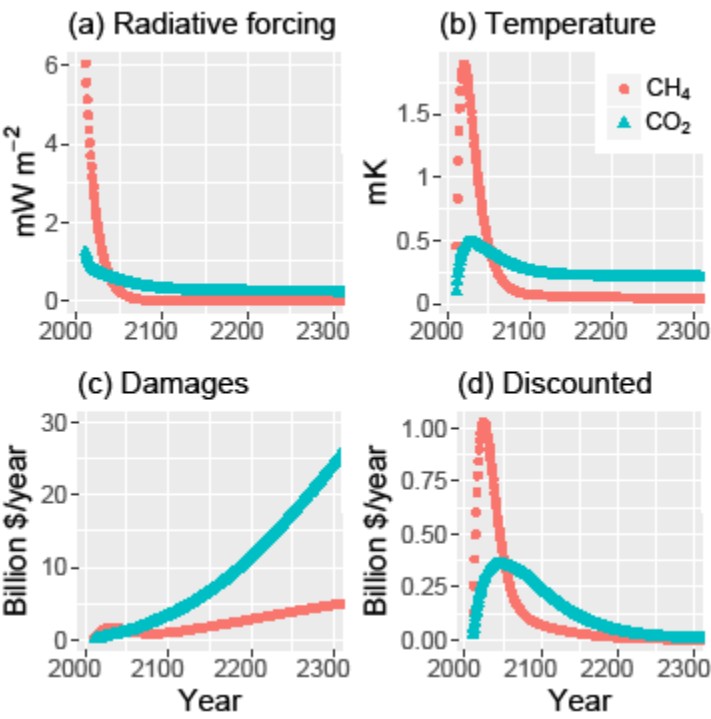

**Figure 1: Impacts of emission pulses of CH₄ and CO₂. Radiative forcing (a), temperature(b), damages (c), and discounted damages**
10  **(3%, d) for an emission pulse of 28.3 MT CH₄ (10 ppb in the first year) and 24.8 times as much CO₂ emissions by mass. The underlying scenario is RCP6.0, with other parameters at their central values.**

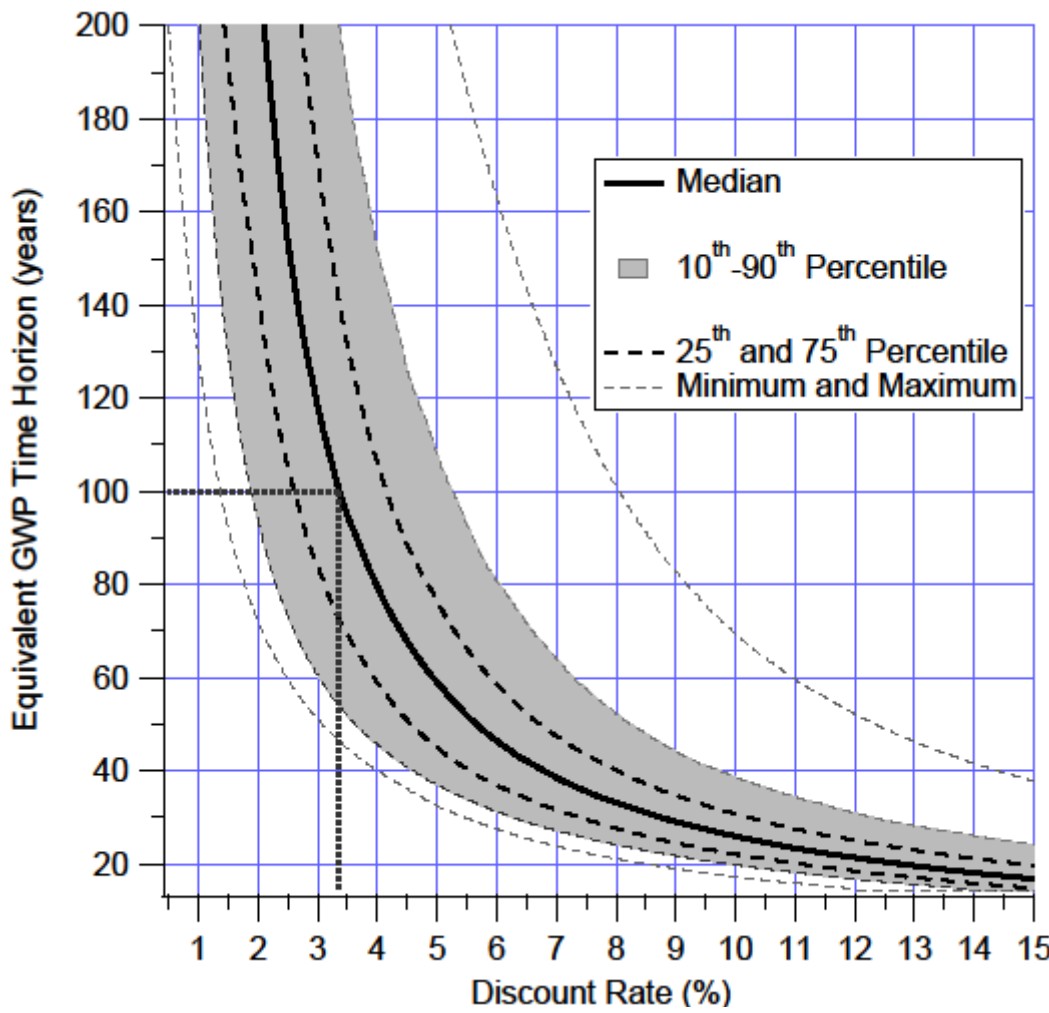

**Figure 2: GWP timescales consistent with discount rates based on consistency of the GWP ratio with the ratio of net present damages of $CH_4$ and $CO_2$, including the interquartile and interdecile bands and maximum and minimum values based on a sensitivity analysis.**

| Parameter | Ratio of highest to lowest damages estimate |
|---|---|
| GDP | 2.07 |
| Damage exponent | 1.63 |
| Scenario | 1.31 |
| Temperature offset | 1.26 |
| Climate sensitivity | 1.16 |
| Forcing imbalance | 1.02 |

**Table 1: Parameter sensitivity analysis: Examining the sensitivity of the GWP-discount rate equivalency as shown in the uncertainty ranges in Fig. 2 as a function of the individual parameters of the calculation. The ratio is calculated as the ratio of the median of the estimated GWPs given the highest and lowest value of each parameter. Results in this table are derived assuming a discount rate of 3%.**

| Gas | Lifetime | Optimal timescale | $GWP_{100}$/damage ratio | $GWP_{20}$/damage ratio |
|---|---|---|---|---|
| $CH_4$ | 12.4 | 120 (84-172) | 1.15 (1.52-0.87) | 3.4 (4.49-2.57) |
| $N_2O$ | 121 | *52 | 0.85 | 0.84 |
| HFC-134a | 13.4 | 115 | 1.11 | 3.2 |
| HFC-23 | 222 | *105 | 0.71 | 0.62 |
| PFC-14 | 50000 | >400 | 0.62 | 0.45 |

**Table 2: Optimal timescale of non-$CO_2$ gases. Implicit timescale evaluated for non-$CO_2$ gases with the GWP to damage ratio for the two most common GWP timescales. Asterisks indicate no exact match between GWP ratio and damage ratio, closest value is given instead. The third and fourth columns show the ratio of the GWP for a given gas to the calculated damage ratio. Interquartile uncertainty ranges are presented for the timescale and damage ratios for $CH_4$. Results in this table are derived assuming a discount rate of 3%.**