# Peer review of "A quantitative approach to evaluating the GWP timescale through implicit discount rates"

_Earth System Dynamics, 2018_

## Referee Comment (RC1) · W.J. Collins (Referee) · 2 Mar 2018

This paper usefully links the well-used GWP metric with more economic comparisons of the ratios of damages. It is suitable for publication and I only have minor comments.

The relationship to the Paris goals are only briefly alluded to. I suggest including a longer discussion of the differences between: a temperature limit, a long-term temperature goal, and least economic cost. Presumably neither a temperature limit nor a long-term temperature goal are optimal economically using the damage function here? Is this a problem with the Paris agreement? The choice of metric depends entirely on the choice of target, and the authors here are implicitly assuming that least eco-

nomic damage is the most important target. The authors dismiss short-term GTPs as implying unrealistic discount rates, which of course they do for an economic damage target. However, if the Paris agreement is taken to imply that temperatures should not be allowed to exceed 2 degrees, then a GTP with a time horizon ending at the time of peak warming (20-30 years) is an entirely appropriate metric. Similarly, if the Paris agreement is taken to mean a long-term goal to stabilise at 2 degrees then GWP* is the appropriate metric.

The overall formula for the damage function needs to be shown as a function of temperature, discount rate, GDP etc.

Page 1, line 26: Maybe a different word other than "endpoint" could be used so as to avoid confusion with the later discussion of integrated and endpoint metrics.

Page 2, first paragraph: The main difference between GTP and GWP is the difference between endpoint and integrated metrics. This should be brought out more in this paragraph. The iGTP could be mentioned as it is more similar to GWP than GTP.

Page 2, second paragraph: Boucher ESD 2012 should also be discussed for economically-based equivalences.

Page 4, line 1. These GDP pathways should be shown (maybe in the supplement).

Page 4, line 6. It is not obvious why 1951-80 should be chosen as a baseline. A problem with damage functions that are non-linear functions of temperature is that a point needs to be chosen when temperatures were optimal.

Figure 1: I was surprised by the shape of 1 (c). Why does the damage from CH4 keep increasing? Is the damage an integral quantity, or is this increase purely due to an exponential increase in GDP? In 1(d) the damage decreases. Is this because the discount rate is larger than the GDP growth? With the GDP growth of 2.06% would a discount rate of less than 2% give an increasing damage for a gas like CO2 with a non-decaying component?

Page 6, line 13: I don't think "exponential function of temperature" is the right term for temperature raised to a power.

Table 2: The ranges (either 25%-75% or 10%-90%) need to be shown as well as the central value. These are quite large for the timescales and may well include 1.0 for many of the damage ratios.

Page 6, line 25: GWP100 seems to agree very well with the 3% discount rate within the uncertainty rather than overvaluing or undervaluing.

Page 6, line 25-29: I didn't understand this sentence. Are you saying that the uncertainty in GWP100 is such that it covers agreement with the 3% discount? If so, that seems to contradict the previous sentence which suggested a under/overvaluing.

---

## Referee Comment (RC2) · Anonymous Referee #2 · 9 Mar 2018

The paper provides an interesting analysis on the connection between GWP timescale and discounting rates. However, I think that before publication, some major issues need to be addressed.

First, while the paper acknowledges a lot of recent articles that discuss GWP, it does not adequately discuss recent articles that look at climate metrics in an economic framework (such as Tol et al. 2012 and Mallapragada and Mignone 2017). There needs to be more incorporation of these types of studies to show how this study builds on the existing literature.

Second, the authors seem to misunderstand the messages of several recent studies,

such as Shindell et al. 2017 and Ocko et al. 2017. These studies are not advocating for a shorter time horizon for GWP, as this paper implies in both the main text and the supplemental information. Rather, they are advocating for using BOTH short AND long-term time horizons to capture the full scope of climate impacts over all timescales – a key distinction that is not depicted in the text. The paper in its current forms criticizes these studies for something that they are not doing. Further, the authors frame their motivation around the fact that studies are advocating GWP20 to then show that GWP100 fits better with discount rates, but because these studies are not simply advocating GWP20, it makes the authors appear naïve to the existing literature. Further, there is a strong reason behind why other timescales are not promoted which needs to be acknowledged (it is not simply a lack of quantification in research efforts) – that just as it is difficult to move the policy community away from the comfortable GWP, it is reasonable to believe that it will be equally as difficult to move the community away from 20 and 100 year timescales of which they are also most familiar with.

Third, it would be great if the damages function description went into more details about what is included in "damages." For example, I believe the authors make it clear later on that health or agriculture impacts from methane were not included. So what is included? Those damages are part of what makes near-term impacts so important to reduce, which justifies the use of a shorter time horizon.

Finally, I wonder about the argument that we should select a time horizon based off of appropriate discount rates. What if the GWP timescale tells us which discount rates are more appropriate? Why is it necessarily the other way around? The literature on appropriate discount rates is vast and its value is debated as much as GWP timescale selection. The paper makes it seem like there is solid agreement on appropriate discount rates but not GWP timescales, but both are subject to similar challenges and debates.

Minor comments:

1.26: Key criticisms also include the reliance of GWP value on a specified time horizon (that is a value judgement) (e.g. Ocko et al. 2017) and that emissions are not continuous (Alvarez et al. 2012). Would also include citations for each point of criticism that you mention. http://www.pnas.org/content/109/17/6435

2.1: Definitely one of the reasons, stronger than "likely."

2.3: Please explain upfront *why* you assess the choice of time horizon – as it wasn't even listed in your list of criticisms other than in reference to discounting (and it is problematic aside from discounting as well).

2.3: 100 year was also selected as middle ground from IPCC FAR as values for 20, 100, and 500 years were given.

2.8: Not sure why the word "therefore" is here. A description of why 100 year was selected does not in itself provide justification for why scientists are promoting 20 years. It is because 100 years does not adequately capture near-term impacts as it masks the importance of short-lived climate pollutants in the near-term. There needs to be a better transition from the 100 year discussion to the 20 year discussion.

2.10: Papers such as Ocko et al. 2017 are not pushing for shortened time horizon, they are pushing for a two-valued GWP metric that includes BOTH 20 and 100 year time horizons. Very important distinction that needs to be clarified, as there are efforts (some livestock groups) that push for short time horizon only.

2.13: Part of the reason that other timescales are not suggested is because of the climate policy community's familiarity with 20 and 100 years. Just as they don't want to adopt a whole new metric, it is very plausible that they will reject a new time horizon. Since 20 and 100 years are adequate for near- and long-term, pushing for say 30 and 200 year time horizons may be counter-productive.

2:20: There are more recent papers that need to be cited that look at the intersection of climate metrics and economics (Tol et al. 2012; Mallapragada and

Mignone 2017). http://iopscience.iop.org/article/10.1088/1748-9326/7/4/044006/meta http://iopscience.iop.org/article/10.1088/1748-9326/aa7397

2.31: Why CO2 and CH4 only? Justification needed, such as represent the largest long-lived and short-lived climate pollutant contributor's to today's radiative forcing.

3.5: Why is a pulse of 28.3 Mt of CH4 used, just bc of 10ppb? Why not today's annual emissions of methane from human activities (around 300-400 Mt)?

3.12: What radiative efficiencies are used? Should specify this since you go into so much detail of other parameter values. I'm assuming radiative efficiencies are from IPCC AR5 but as you cite in your references, there are more recent calculations in Etminan et al. 2016.

3:28: What damages are included by using this function?

3:30: Please include citations for the first alternative.

4.2: Suggest mentioning how these results fit in with scientific literature that has looked at these tradeoffs for decades.

---

## Referee Comment (RC3) · Anonymous Referee #3 · 12 Mar 2018

Review of "A quantitative approach to evaluating the GWP timescale through implicit discount rates"

I am not an expert in "Economics of climate change" and "discount rates" applied in estimating damages from climate change. Hence, I ask the editor to rely on the opinion of reviewers who are experts in assessing the economic damage from climate change. Here I am providing just a couple of minor comments.

A recent paper (Modak, A., G. Bala, K. Caldeira, and L. Cao, 2018: Does shortwave absorption by Methane influence its effectiveness? Climate Dynamics, https://doi.org/10.1007/s00382-018-4102-x) shows that the efficacy of methane forc-

ing is only 80% relative to CO2 forcing. The lower efficacy affects the estimation of GTP and hence the damages. What is the implication of this result to the conclusion reached in your study? Discuss.

In the abstract and in the 2nd paragraph of the Introduction section, it is stated that GWP assumes constant future concentrations. I believe this is true only for the baseline state. GWP is estimated for a case where the concentration of the gases decay with time. The integrated radiave forcing is calculated for the time evolving concentrations relative the baseline.

Fig. 1c and d: Is the unit for the damages and discounted damages correct? Should it be Billion $ per year? Same issue for Fig. S1

─────────────────────────

---

## Referee Comment (RC4) · Anonymous Referee #4 · 12 Mar 2018

The manuscript makes a useful contribution to the literature by exploring explicitly how different time scales for GWP relate to GHG equivalence ratios based on damage costs and different discount rates. It is clearly written and highly readable. I have no fundamental concern with the technical approach and quantitative results, but I feel the manuscript needs to work a bit harder to develop its value proposition, discussion of results including sensitivity analysis, and finally the conclusions, before it is fit for publication.

I'm comfortable with and largely endorse the comments already posted by Bill Collins and anonymous referee #2, and will try not to repeat the specific points they made.

[Figure]

My main concerns where I feel the manuscript needs to work harder are as follows:

1) value proposition: it is mainly in the SI that the authors acknowledge prior work that linked GHG equivalencies based on damage costs and discount rates to GWP. I believe this needs to be brought into the main paper up-front, and the authors need to do a better job explaining where their study adds value to those existing studies. For example, one could argue that their approach is simply a reverse reading of Boucher (2012). I don't think that accusation would be justified, but neither is it justifiable for the manuscript not to recognise the fact that a range of studies have already found that discount rates around 2-3% give the same GHG equivalence between CH4 and CO2 as GWP100. In this context, in the discussion, I would have liked to see a better explanation why their GWP100-equivalent discount rate of 3.3% is higher than that derived by both Boucher and Fuglestvedt et al.

2) discussion of results including sensitivity analysis: in my view, the authors should include an explicit simulation of results if climate-carbon cycle feedbacks following a pulse emission of CH4 are included. The IPCC AR5 and subsequent studies demonstrated that including this results in a significant increase in the GWP100. This is flagged (p7 of the manuscript) but appears not to have been included in the actual sensitivity analysis. It should be fairly easy to modify the radiative forcing calculations to simulate climate-carbon cycle feedbacks and it doesn't have to change the study design at all. There really is no good justification in my view not to include this, other than this is not how the GWP has been defined historically – but from a scientific consistency perspective, it makes no sense to include an effect for one gas (CO2) but not for the other. Including this in the sensitivity analysis (perhaps as a special case, since this is a binary choice rather than something that can be expressed via a pdf) would at least tell us how important this is when we are concerned about choosing GHG equivalencies based on damage functions and discount rates. I could even live with the authors running this only for a central estimate for all other parameters so we can get an order-of-magnitude sense.

[Figure]

Related to this, but more difficult to do (hence I would not insist that this is done quanti-tatively) is consideration of the rate of change as a source of damages. Again this could be parameterised and quantified, but there is a large degree of arbitrariness how much weight to place on rate of change vs amount of change. The manuscript would be much stronger though if it could demonstrate under what circumstances including the rate of change might affect the conclusions, or whether the conclusions might be robust even if rate of change damages have been incorporated within reasonable bounds.

3) interpretation and conclusions: I would endorse some of the comments made by anonymous reviewer #2, that the authors are effectively beating up a strawman. Yes some people have argued that we should simply 'switch' to GWP20, but the more intel-ligent arguments are all for considering the effect of multiple alternative time horizons to inform abatement decisions and policy choices. See e.g. the conclusions in Levasseur et al 2016 (doi: 10.1016/j.ecolind.2016.06.049) regarding the use of multiple time hori-zons and metrics in lifecycle assessment. The discussion and conclusions need to add quite a bit of nuance to reflect what those studies actually say, and hence the degree to which this manuscript challenges their conclusions or simply adds another dimension that can help choosing the right metric for the right purpose.

There are two additional points that the discussion and conclusion needs to address:

(a) one is that a key context in which GHG metrics are used are in emission trading schemes, and to help governments evaluate policy choices that directly affect near-term commercial decisions, i.e. policy that would "alter the use of capital in the private sector". So there are very different contexts in which GHG metrics are actually used in climate policy and where different discount rates are commonly applied, and the paper would be stronger and more relevant if it recognised and addressed these explicitly.

(b) The second is a recognition that IAMs used to design cost-minimising emission pathways often use a discount rate of 5%. Given that a(nother) key use of GHG metrics is to help IAMs make trade-offs between different gases with different mitigation costs,

this should enter into the discussion in this paper. I don't think this materially changes the conclusions since we know that different GHG metrics don't have a massive effect on total mitigation costs (although there is a systematic effect especially when moving towards GWP20), but the issue is not trivial especially for countries or sectors with non-negligible non-CO2/SLCF sources. Some discussion on this is needed.

I believe that all the above points (with the exception of quantifying the effect of including climate-carbon cycle feedbacks for CH4) can be addressed by a careful revision of the text itself. The manuscript needs to avoid what currently appears as the too-simplistic conclusion that "actually, GWP100 is largely ok, let's move on" (which is how I read P8L22). The fundamental finding from virtually all metrics papers is that the right metric depends on the application, and hence it is rather jarring to read a conclusion that continued use of GWP100 is 'reasonable' without any caveat.

I am not repeating the above points in my specific comments below and would be happy for the authors to decide how they can best address them.

Specific comments:

P1L22: insert 'emission' after gases – we're talking about emission metrics here

P2L3: 'endorsed' is too strong in my view for the UNFCCC – 'used' is more factual, I cannot recall an explicit endorsement in the sense that the UNFCCC would have explained and justified its choice.

P2L11: I believe the correct term for GTP is Global Temperature CHANGE Potential

P2L12: the reason why GTP downplays SLCFs is not primarily that it is temperature based but that it is a point metric. iGTP is very similar to GWP.

P2L22-27: editorial only: I prefer if introductions don't include the conclusions but rather focus on making the point of why the conclusions are worth having.

P3L15: shouldn't the N2O effect on CH4 forcing depend on the RCP pathway? Perhaps this was done but this isn't clear to me from the text.

P4L9: 'future years are cooler than present': helpful if you could indicate what years we are talking about (presumably you mean after 2200 or thereabouts, depending on the reference period/warming – meaning that much of those will be heavily discounted anyway).

P4L21: here and later, please clarify where you truncate your damage calculations (when I read this sentence, I thought you truncate at 2300, but later (P5L14) it seems you truncate at 2500). You note below that this may matter for very low discount rates. Can you quantify/illustrate this?

P6L4: I think the entire sensitivity discussion should note that projecting damages multiple centuries into the future is increasingly fraught with difficulties. The AR5 chose not to evaluate GWP500 because the authors felt that (deep) uncertainties were simply too large – but here you evaluate damages from temperature responses from forcing 500 years into the future? At least a comment on this is needed here – the discussion of what percentage of total damages occurs up to a given year for CH4 and CO2 is useful in this context and could be linked to this point about uncertainty.

P8L2: I feel the statement "We note no metric designed to tradeoff emission pulses is consistent with stabilization" is too strong. Of course, no metric in itself delivers stabilisation, but almost any metric can be used wisely enough to help countries achieve stabilisation.

---

## Author Comment (AC1) · 19 Apr 2018

This paper usefully links the well-used GWP metric with more economic comparisons of the ratios of damages. It is suitable for publication and I only have minor comments.

We would like to thank the referee for their comments. For responses comment by comment, see below. All author replies are in red. There is also a summary of new sensitivity analyses that is included at the end of this comment reply.

The relationship to the Paris goals are only briefly alluded to. I suggest including a longer discussion of the differences between: a temperature limit, a long-term temperature goal, and least economic cost. Presumably neither a temperature limit nor a long-term temperature goal are optimal economically using the damage function here? Is this a problem with the Paris agreement? The choice of metric depends entirely on the choice of target, and the authors here are implicitly assuming that least economic damage is the most important target. The authors dismiss short-term GTPs as implying unrealistic discount rates, which of course they do for an economic damage target. However, if the Paris agreement is taken to imply that temperatures should not be allowed to exceed 2 degrees, then a GTP with a time horizon ending at the time of peak warming (20-30 years) is an entirely appropriate metric. Similarly, if the Paris agreement is taken to mean a long-term goal to stabilise at 2 degrees then GWP* is the appropriate metric.

The reviewer brings up an interesting question regarding the goals of global climate agreements. To some extent, the optimal design of a global goal is beyond the scope of this manuscript, but given that one important use of climate metrics is within cap and trade programs that are meant to help achieve these global goals, some discussion would be appropriate. To that end, we propose to bring some of the text that was originally in the SI into the main manuscript, with some updating and editing. Such language might look like the following:

This paper does not address the construction of global climate goals (such as the 2 °C goal in the Paris agreement). However, because one use of metrics is to enable trading between gases in cap and trade programs, and because cap and trade programs are one of the measures used to help achieve these long-term targets, a discussion of the relationship between metrics and temperature stabilization is relevant. A number of authors have recognized that the GWP is not designed to achieve stabilization goals (Sarofim et al. 2005, Smith et al. 2012, Allen et al. 2016). Some actors (Brazil INDC, 2015) have claimed that certain metrics such as the Global Temperature Potential (Shine et al. 2005) or the Climate Tipping Potential (Jorgensen et al. 2014) are more compatible with a stabilization target such as 2 degrees C because they are temperature based. However, these metrics are also not designed to achieve stabilization goals, but rather to achieve a temperature target in a single given year. The challenge is that in any year after stabilization, any trading between emission pulses of carbon dioxide and a shorter-lived gas will cause a deviation from stabilization. For example, trading a reduction in methane emissions for a pulse of $CO_2$ emissions will lead to a near term decrease in temperature, but also a long-term increase in temperature above the original stabilization level. One solution to the problem is a physically-based one. Allen et al. (2016) suggest trading an emission pulse of carbon dioxide against a sustained change in the emissions of a short-lived climate forcer. This resolves the issue of trading what is effectively a permanent temperature change for a transient one. However, the challenge becomes one of implementation, as current policy structures are not designed for addressing indefinite sustained mitigation. Alternatively, a number of researchers (Daniel 2012, Jackson 2009, Smith et al. 2012) have suggested addressing $CO_2$

mitigation separately from short-lived gases. Such a separation recognizes the value of the cumulative carbon concept in setting GHG mitigation policy (Zickfeld et al. 2009).

We note as an aside that for a purely economic approach to recommend an absolute target such as 2 degrees would require a world in which damages were infinite above 2 degrees, and zero below two degrees (see e.g., the discussion of "cost-effectiveness" in Deuber et al. 2013, Physico-economic evaluation of climate metrics: A conceptual framework).

The overall formula for the damage function needs to be shown as a function of temperature, discount rate, GDP etc.

We propose to add text such as

Damages: Damages as a percent of GDP were calculated by multiplying a constant times the temperature squared. E.g., $D(2050) = a*\Delta T(2050)^2*GDP$, and the net present value is calculated using the discount rate such that $NPV(D(t)) = D(t)/(1+r)^{t-2010}$.

Page 1, line 26: Maybe a different word other than "endpoint" could be used so as to avoid confusion with the later discussion of integrated and endpoint metrics.

We will replace endpoint here with "measure of impact"

Page 2, first paragraph: The main difference between GTP and GWP is the difference between endpoint and integrated metrics. This should be brought out more in this paragraph. The iGTP could be mentioned as it is more similar to GWP than GTP.

We propose to add a discussion along the lines of the following:

While we argue that quantitative justifications for timescales within the GWP are rare, as reflected by the judgment of the IPCC authors that no scientific arguments exist for selecting given timescales, there is a rich literature addressing many aspects of climate metrics. Deuber et al. (2013) presents a conceptual framework for evaluating climate metrics, laying out the different choices involved in choosing the measure of impact of radiative forcing, temperature, or damages, and temporal weighting functions that can be integrative (whether discounted or time-horizon based) or based on single future time points. Deuber et al. conclude that the Global Damage Potential (GDP) could be considered a "first-best benchmark metric", but recognize that the time-horizon based GWP has advantages based on limited value-based judgments to a choice of time horizon, reducing scientific uncertainty by limiting the calculations of atmospheric effects to radiative forcing, and eliminating scenario uncertainty by assuming constant background concentrations. Mallapragada and Mignone (2017) present a similar framework which also notes that metrics can consider a single pulse of a stream of pulses over multiple years. Several authors have noted that under certain simplifying assumptions, the GWP is equivalent to the integrated GTP, and therefore any timescale arguments that apply to analyses of one metric would also apply to the other (Sarofim GTP paper and related references).

Page 2, second paragraph: Boucher ESD 2012 should also be discussed for economically-based equivalences.

We propose to bring a discussion of Boucher 2012, Fuglesvedt 2003, Reilly 2001, and other relevant papers into a paragraph in the introduction. There is some relevant language that was originally in the SI that, with some modifications, might serve:

> We acknowledge that a number of authors have considered the use of relative damages as a potential metric, and in doing so have often compared the resulting global damage potential to the GWP. However, we believe that none of these papers has done so with the explicit goal of evaluating GWP timescales, and determining the discount rate implied by any given timescale choice. Boucher (2012) in particular compares a global damage potential to the $GWP_{100}$ with an uncertainty analysis, and finds, similar to our manuscript, that the median values of each approach are consistent with one another. Fuglestvedt et al. (2003), in an assessment of a number of different approaches to metrics, performed a similar calculation, and also found that a 100 year GWP was consistent with a discount rate on the order of 2%. De Cara et al. (2005), in an unpublished manuscript, calculated the relationship between discount rates and time horizon, though they assumed linear damages.

Page 4, line 1. These GDP pathways should be shown (maybe in the supplement).

We will include a graph of the GDP pathways in the SI.

Page 4, line 6. It is not obvious why 1951-80 should be chosen as a baseline. A problem with damage functions that are non-linear functions of temperature is that a point needs to be chosen when temperatures were optimal.

We agree that there is no clear best baseline: that is why we did a sensitivity analysis using a baseline of 0 (effectively, that damages are a function of temperature change relative to 2010) and a baseline of 0.8 degrees (assuming that damages are a function of temperature change since preindustrial times) were used. Table 1 indicates that this choice can have a difference of up to 26% in terms of damages. We propose to add a sentence on this sensitivity analysis to the paper.

Figure 1: I was surprised by the shape of 1 (c). Why does the damage from CH4 keep increasing? Is the damage an integral quantity, or is this increase purely due to an exponential increase in GDP? In 1(d) the damage decreases. Is this because the discount rate is larger than the GDP growth? With the GDP growth of 2.06% would a discount rate of less than 2% give an increasing damage for a gas like CO2 with a non-decaying component?

We propose to add a sentence discussing this effect:

> Fig. 1c demonstrates the dramatic increase in damage over time due to the relationship of damage to economic growth. In the case of $CH_4$, damage peaks in 2032 and declines until 2080 as a result of the short lifetime of the gas. The increase in damages after 2080 is due to the component of the temperature response function that includes a 409 year timescale decay rate,

such that after 100 years the decrease in the $\Delta T^2$ is about 0.5%, and because that decay rate is slower than the rate of consumption growth, the net damages grow.

Page 6, line 13: I don't think "exponential function of temperature" is the right term for temperature raised to a power.

Will change to "polynomial function".

Table 2: The ranges (either 25%-75% or 10%-90%) need to be shown as well as the central value. These are quite large for the timescales and may well include 1.0 for many of the damage ratios.

We propose to update the table as follows, including uncertainty ranges for $CH_4$: Uncertainty ranges for the longer-lived gases are more challenging, as the non-monotonicity adds complications.

**Table 1: Parameter sensitivity analysis: Examining the sensitivity of the GWP-discount rate equivalency as shown in the uncertainty ranges in Fig. 2 as a function of the individual parameters of the calculation. The ratio is calculated as the ratio of the median of the estimated GWPs given the highest and lowest value of each parameter. Results in this table are derived assuming as discount rate of 3%.**

| Gas | Lifetime | Optimal timescale | $GWP_{100}$/damage ratio | $GWP_{20}$/damage ratio |
|---|---|---|---|---|
| $CH_4$ | 12.4 | 120 (84-172) | 1.15 (1.52-0.87) | 3.4 (4.49-2.57) |
| $N_2O$ | 121 | *52 | 0.85 | 0.84 |
| HFC-134a | 13.4 | 115 | 1.11 | 3.2 |
| HFC-23 | 222 | *105 | 0.71 | 0.62 |
| PFC-14 | 50000 | >400 | 0.62 | 0.45 |

**Table 2: Optimal timescale of non-$CO_2$ gases. Implicit timescale evaluated for non-$CO_2$ gases with the GWP to damage ratio for the two most common GWP timescales. Asterisks indicate no exact match between GWP ratio and damage ratio, closest value is given instead. The third and fourth columns show the ratio of the GWP for a given gas to the calculated damage ratio. Interquartile uncertainty ranges are presented for the timescale and damage ratios for $CH_4$. Results in this table are derived assuming as discount rate of 3%.**

Page 6, line 25: GWP100 seems to agree very well with the 3% discount rate within the uncertainty rather than overvaluing or undervaluing.

We propose to modify the text "However, the analysis shows that the 100-year timescale is consistent, within the interquartile range, with the 3% discount rate that is commonly used for climate change analysis."

Page 6, line 25-29: I didn't understand this sentence. Are you saying that the uncertainty in GWP100 is such that it covers agreement with the 3% discount? If so, that seems to contradict the previous sentence which suggested a under/overvaluing

We chose to delete the sentence in question. Upon reflection, the uncertainty in the GWP100 is due to factors like uncertainty in radiative efficiency, but (as shown in a sensitivity calculation) the timescale is not sensitive to radiative efficiency uncertainty.

Additional Sensitivity Analyses in response to various Referee Comments:
The four referees raised a number of points, several of which were inter-related. In response to these points, we have performed a number of additional sensitivity analyses, which will be detailed here. One lesson from these analyses: the analysis is robust to a surprising number of changes. This relates to the fact that any change in the analysis which changes both the GWP and the damage ratio generally cancels out in terms of calculating the relationship between discount rate and timescale: e.g., most factors in the causal chain from emissions through radiative forcing. That includes the size of the emissions pulse, the radiative efficiency of the gas, and the lifetime of the gas (within certain limits). Note that changes in these parameters may well change the best estimate of the relative importance of reducing methane compared to $CO_2$, but they do not change the implicit timescale that is the focus of this paper.

In contrast, anything which changes only the damage ratio can have a larger impact on the implicit timescale. This includes the factors which have the largest influence on the uncertainty, as reflected in Table 1: the rate of GDP growth, the damage exponent, the scenario (because the GWP assumes constant concentrations), the baseline temperature from which damages are calculated, and the climate sensitivity.

A third category is influences which are specific to a given gas, whether the efficacy of that gas (e.g., Modak et al. noted by Referee #3), or the health effects of methane-related ozone productions. Like changes to radiative efficiency or lifetime, these influences can also change the best estimate of the relative importance of reducing methane compared to $CO_2$, but because they are specific to a single substance and not generalizable between substances, they would not be appropriate for calculating implicit timescales.

Herein, we summarize the additional sensitivity analyses:

Pulse Size:

In response to a comment by Referee #2, we performed a sensitivity analysis on the size of the emissions pulse. Using 373 MMT (one year's emissions according to Saunois 2016, http://iopscience.iop.org/article/10.1088/1748-9326/11/12/120207, where global emissions are 559 MT of which 2/3rds are anthropogenic), the relative damage ratios of $CH_4/CO_2$ are the following:

| quartile | 25% | 50% | 75% |
|---|---|---|---|
| Damage Ratio | 18.80 | 25.05 | 32.69 |

A similar analysis for the 28.3 MMT used in the paper, performed in response to a comment by reviewer 1, yields

| quartile | 25% | 50% | 75% |
|---|---|---|---|
| Damage Ratio | 18.84 | 25.12 | 32.86 |

The differences between the two analyses are less than 1%. We will note this sensitivity analysis in the paper. Arguments can be made for either the larger quantity or the smaller quantity depending on the purpose of the analysis. When GWPs are used to inform decision making, these decisions are often regarding mitigation actions at the national or sub-national scale, which is closer to the 28 MMT scale than the 370 MMT scale. When GWPs are used to compare $CO_2$ equivalents for global scenarios, then the global annual number might be more appropriate. But in any case, this appears to be an uncertainty much smaller than any of the others considered in this paper.

Radiative Efficiency:

Referee #2 also raised a question regarding the use of the IPCC AR5 radiative efficiencies rather than the more recent Etminan et al. 2016 article. In order to test the general sensitivity of this approach to changes in the radiative efficiency of methane, a sensitivity analysis was performed, wherein forcing from $CH_4$ was doubled compared to the standard calculation. Relative damages of $CH_4$ and $CO_2$, as might be expected, also doubled:

| quartile | 25% | 50% | 75% |
|---|---|---|---|
| Damage Ratio | 37.98 | 50.35 | 65.50 |

However, calculated GWPs based on the updated radiative forcing also double: the new GWP100 being 56.79, and the new GWP20 being 166.92. The optimal timescale using the central parameters ends up being 119.79 years, under 1/10th of a percent different than the original calculated timescale of 119.85 years.

We can also mention this in the paper.

Consistent Climate Carbon Feedbacks:

Referee #4 points out, with good cause, that it would be a good sensitivity analysis to calculate an implicit timescale using consistent assumptions about climate-carbon feedbacks for both carbon and methane lifetimes (whereas, in the main body of the paper, we used a $CO_2$ lifetime that included climate-carbon feedbacks, and a methane lifetime that did not, consistent with the approach of the IPCC AR4 and the GWP values reported in the main body of chapter 8 in IPCC AR5). Using the "no-climate-carbon-feedback" $CO_2$ equation from Gasser et al. 2017, we can calculate $CH_4/CO_2$ damage ratios:

| quartile | 25% | 50% | 75% |
|---|---|---|---|
| Damage Ratio | 20.57 | 27.16 | 35.23 |

This can be compared to the original damage ratios from the paper:

| quartile | 25% | 50% | 75% |
|---|---|---|---|
| Damage Ratio | 18.84 | 25.12 | 32.86 |

The damage ratios are about 8% larger under the consistent no-CC-feedback case than in the $CO_2$-feedback/$CH_4$-nofeedback case, due to what is now a shorter $CO_2$ lifetime. But a GWP calculated with consistent assumptions regarding climate-carbon feedbacks also changes. The no-feedback GWP100 is now 30.74 rather than 28.64, a 7% increase. These two effects largely cancel out, leaving the effect on implicit timescales smaller, again, than most of the other sensitivities examined in the paper.

| quartile | 25% | 50% | 75% |
|---|---|---|---|
| consistent no cc feedback timescales | 170.56 | 117.9 | 83.38 |
| original analysis | 170.87 | 118.30 | 83.81 |

So while the damage-ratio differences are on the order of 8%, the implicit timescale differences are less than 1%. Similar to the impact of changing methane's radiative forcing, a large impact on damage ratios has a small impact on implicit timescales because the update happens in both the calculation of the damage-ratio AND in the calculation of the GWP, and therefore cancels out to a large extent.

We show here a calculation that was consistent in not including climate carbon feedbacks in either damage calculation. While it would be possible to do a similar consistent calculation with climate carbon feedbacks in both damage calculations, implementation of the climate-carbon feedback for methane is not without challenges. Such a calculation requires first calculating of the temperature impact resulting from an emissions pulse over the full time period (as done in the original), and then, for the temperature change in each year, requires a calculation of the carbon perturbation resulting from that temperature change over the remainder of the time period, and then calculating the additional forcing and temperature change resulting from that carbon perturbation. Therefore, we decline to do this calculation at this time (absent an already available

methane lifetime equation with the CC feedback built in, which the authors do not have on hand). The authors are already on record as preferring a GWP metric that does not include CC feedbacks in either the $CO_2$ or the $CH_4$ response rather than inclusion of CC feedbacks in both $CO_2$ and $CH_4$ responses – see the response by Sarofim, Giordano, and Crimmins to the Gasser paper for a more detailed explanation (https://www.earth-syst-dynam-discuss.net/esd-2016-55/esd-2016-55-SC1.pdf)

Ramsey Discounting:
Of all the sensitivity analyses (other than discount rate), the calculation of implicit timescales was most sensitive to changes in the assumption about future GDP growth rates. The original draft raised the question as to what extent this sensitivity to GDP growth might diminish were the discount rate to be a function of GDP, as in a Ramsey discounting framework. In this case, a high GDP growth rate (which implies large future damages, and therefore a low methane/CO2 damage ratio) would be counteracted by a high discount rate which would be expected to lead to a high methane/CO2 damage ratio.

For this sensitivity analysis, the Ramsey discounting approach was calibrated to yield an average discount rate for the first 30 years of the analysis of 5% for the reference GDP growth rate, with a pure rate of time preference of 0.01% (a very low pure rate of time preference being consistent with the assumption that, holding consumption constant, all generations should be given equal weight). That required an elasticity of marginal utility of consumption of 1.53.

Interestingly, the ratio of the median damage estimate from the low GDP scenario to the high GDP scenario was still about a factor of 2, except that now the low GDP scenario leads to the lowest damage ratio and the high GDP scenario leads to the highest damage ratio: effectively, in the Ramsey case, the discount rate effect overwhelms the GDP growth rate effect. Note that the equivalent timescale in the median GDPref case under this set of assumptions is 135 years.

| GDP scenario | | GDPlow | GDPref | GDPhigh |
|---|---|---|---|---|
| Discount rate, first 30 years-> | | 2.5% | 5% | 7.5% |
| Discount rate, full run-> | | 0.8% | 1.5% | 2.3% |
| Ramsey discounting approach | 25% | 11.28 | 15.71 | 21.78 |
| | 50% | 16.08 | 22.72 | 31.05 |
| | 75% | 18.42 | 26.40 | 35.16 |
| | | 3% | 3% | 3% |
| Original constant discount rate approach | 25% | 29.96 | 21.11 | 14.46 |
| | 50% | 38.69 | 27.58 | 18.70 |
| | 75% | 42.52 | 30.29 | 19.81 |

Rate of Change Calculation:
Referee #4 also brought up the idea of adding in damages from rate of change of temperature as well as from absolute temperature change. This was an important addition in, for example, Manne & Richels (2001) where it turned a dynamic optimization metric from one that didn't

value methane until a decade or two before threshold temperatures were reached to a metric which had fairly constant value over time.

In order to test this, we added code that included damages from rate of change. The damage is calculated by taking the square of the rate of change and multiplying by the GDP and a constant – similar to the damage calculation for absolute temperature.

For the first test, peak rate of change damages under the central scenario were calibrated to be equal to the damages in 50 years (2060), where there is 1 degree of temperature change (above the temperature offset) under central parameter estimates. This led to an increase in damage ratio of 2.4%.

A second test, with peak rate of change damages 10 times as large, led to an increase in the damage ratio of 5.1%. That yields a timescale change from 120 years to 112 years.

The small impact of the rate of change damage is likely due to timing of peak rates of change under RCP6. In this analysis, there is an initial high rate of change for the first few years, followed by a secondary (though smaller) peak about 60 years in. Because of the GDP growth in the interim, damages at the 60 year secondary peak are higher than damages in the initial years. Therefore, reducing the rate of temperature growth 60 years into the run is more important than reducing the rate of growth at the beginning of the run, and for this reason, reducing short-lived forcer emissions does not have large advantages over reductions in $CO_2$ emissions.

However, this raises a question about the importance of the scenario assumption. Therefore, we also tried the same exercise with the RCP3PD scenario as the baseline. In RCP3PD, in this analysis, the rate of temperature change is at its highest in the first years of the scenario, which means that the increase in rate of change due to additional radiative forcing in those early years can have a disproportionately large impact, favoring short-lived climate forcers. Note that because RCP3PD cools in later years, we chose to set the damages resulting from a negative rate of change to zero.

In this case, the first rate of change analysis yields an increase in the damage ratio of 53%, and the second analysis yields a damage ratio increase of 85%. This yields a timescale change from 94 years (the timescale for RCP3PD without rate of change damages) to 54 years to 42 years.

This is consistent with some other literature such as Bowerman et al. (2013), which suggests that under stringent $CO_2$ mitigation scenarios, the rate of change peaks in the near future, and therefore reduction of short-lived climate forcers can be particularly valuable – but that in scenarios where $CO_2$ emissions continue to rise in the near term, the rate of change peaks further in the future, and therefore delay in short lived climate forcer mitigation will lead to the greatest reductions in peak rate of change.

However, this particular sensitivity analysis was rather crude. In order to improve it, we would have to develop a better reasoning for choosing the parameters involved (e.g., the damage exponent for rate of change, and the damage constant), but also investigate how well our approach models near-term rate of change compared to more complex models, as that is of importance for this calculation.

---

## Author Comment (AC2) · 19 Apr 2018

The paper provides an interesting analysis on the connection between GWP timescale and discounting rates. However, I think that before publication, some major issues need to be addressed.

We would like to thank the referee for their comments. For responses comment by comment, see below. All author replies are in red. There is also a summary of new sensitivity analyses that is included at the end of the reply to William Collins.

First, while the paper acknowledges a lot of recent articles that discuss GWP, it does not adequately discuss recent articles that look at climate metrics in an economic frame- work (such as Tol et al. 2012 and Mallapragada and Mignone 2017). There needs to be more incorporation of these types of studies to show how this study builds on the existing literature.

We propose to move material that was originally in the SI into the main text, as well as incorporating more references such as the ones you suggest. See the following proposed text addition:

> While we argue that quantitative justifications for timescales within the GWP are rare, as reflected by the judgment of the IPCC authors that no scientific arguments exist for selecting given timescales, there is a rich literature addressing many aspects of climate metrics. Deuber et al. (2013) presents a conceptual framework for evaluating climate metrics, laying out the different choices involved in choosing the measure of impact of radiative forcing, temperature, or damages, and temporal weighting functions that can be integrative (whether discounted or time-horizon based) or based on single future time points. Deuber et al. conclude that the Global Damage Potential (GDP) could be considered a "first-best benchmark metric", but recognize that the time-horizon based GWP has advantages based on limited value-based judgments to a choice of time horizon, reducing scientific uncertainty by limiting the calculations of atmospheric effects to radiative forcing, and eliminating scenario uncertainty by assuming constant background concentrations. Mallapragada and Mignone (2017) present a similar framework which also notes that metrics can consider a single pulse of a stream of pulses over multiple years. Several authors have noted that under certain simplifying assumptions, the GWP is equivalent to the integrated GTP, and therefore any timescale arguments that apply to analyses of one metric would also apply to the other (Sarofim GTP paper and related references).

Second, the authors seem to misunderstand the messages of several recent studies, such as Shindell et al. 2017 and Ocko et al. 2017. These studies are not advocating for a shorter time horizon for GWP, as this paper implies in both the main text and the supplemental information. Rather, they are advocating for using BOTH short AND long-term time horizons to capture the full scope of climate impacts over all timescales – a key distinction that is not depicted in the text. The paper in its current forms criticizes these studies for something that they are not doing. Further, the authors frame their motivation around the fact that studies are advocating GWP20 to then show that GWP100 fits better with discount rates, but because these studies are not simply advocating GWP20, it makes the authors appear naïve to the existing literature. Further, there is a strong reason behind why other timescales are not promoted which needs to be acknowledged (it is not simply a lack of quantification in research efforts) – that just as it is difficult to move the policy community away from the comfortable GWP, it is reasonable to

believe that it will be equally as difficult to move the community away from 20 and 100 year timescales of which they are also most familiar with.

We will add clarification that in some cases authors are often suggesting presentation of short (about 20 years) timescales alongside, not in place of, 100 year timescales. However, we do feel that "promoting more emphasis on shorter time horizons" is an accurate description, as providing both GWP20 and GWP100 (for example) is "more emphasis" on shorter time horizons than just providing GWP100. Presented with values for 2 time horizons, one might also expect a decision-maker to have some probability of choosing to use only the shorter one, or to weigh them equally, which would look like a GWP60 (for comparison, a GWP60 would have an median equivalent discount rate of 5% using our methodology). Meanwhile, some authors (e.g., Howarth) do explicitly state that use of the 20 year GWP would better account for relevant climate impacts than 100 year GWPs, and a number of NGOs have followed suit. We propose adding a sentence to recognize some of these nuances:

> Recently, some researchers and NGOs have recently been promoting more emphasis on shorter time horizons, such as 20 years, which would highlight the role of short-lived climate forcers such as $CH_4$ (Howarth et al. 2011, Edwards and Trancik, 2014, Ocko et al. 2017, Shindell et al. 2017). These studies each have different nuances regarding their recommendations – for example, Ocko et al. (2017) suggest pairing the GWP100 with the GWP20 to reflect both long-term and near-term climate impacts – and therefore there is no simple summary of the policy implications of this body of literature, but it is plausible that more consideration of short-term metrics would result in policy that weights near-term impacts more heavily than would result from consideration of the GWP100 in isolation.

Third, it would be great if the damages function description went into more details about what is included in "damages." For example, I believe the authors make it clear later on that health or agriculture impacts from methane were not included. So what is included? Those damages are part of what makes near-term impacts so important to reduce, which justifies the use of a shorter time horizon.

Damages in this case are only climate-related damages. Inclusion of damages due to, for example, the health impacts of $O_3$ are relevant for policymaking purposes as one of the authors has argued elsewhere (see, e.g., Sarofim, Waldhoff, and Anenberg), but we would argue are not appropriate for the timescale discussion. One support for keeping timescale & non-climate effects separate is that if the timescale is adjusted to increase the value of methane to account for the methane-ozone effects, then it will equally increase the value of HFC-134a which has a similar lifetime: but HFC-134a does not have an equivalent ozone-effect. Therefore, instead, the relative climate effects of gases should be calculated using this timescale approach, and then the value of reducing methane can be increased to account for its ozone effects (and the value of reducing $CO_2$ can be adjusted to account for fertilization and acidification effects).

We propose to add a sentence in the conclusions noting that the timescale question is related to, but separate, from the relative value of abating methane and $CO_2$. Issues like ozone-damages resulting from

methane emissions should be addressed by a post-hoc adjustment, like how the ozone and stratospheric water vapor forcing effects are added to the methane GWP, rather than by using a method which truncates valuation of any climate impacts more than 20 years into the future.

Finally, I wonder about the argument that we should select a time horizon based off of appropriate discount rates. What if the GWP timescale tells us which discount rates are more appropriate? Why is it necessarily the other way around? The literature on appropriate discount rates is vast and its value is debated as much as GWP timescale selection. The paper makes it seem like there is solid agreement on appropriate discount rates but not GWP timescales, but both are subject to similar challenges and debates.

Our analysis can be used in either direction, as is discussed in the paper:

> Here we focus on what discount rates are consistent with a GWP time horizon in order to show the discount rates implied by common choices of GWP timescales. The converse calculation is relevant for an audience that has a preferred discount rate and is interested in the implied GWP timescale.

While we agree that there is no single consensus on an appropriate discount rate, we do think that that the framing of discount rates is good way to formally demonstrate the implications of GWP timescales for valuation over time. Part of the impetus of this paper was due to the possibility that some proponents of shorter timescales might not recognize the implicit discount rate embodied in the timescale choice, and might not consider a high discount rate to be desirable. We recognize that the discount rate implications calculated in this paper are a result of many decisions regarding parameters and analytical approaches, and that other approaches might yield different results, but we think that this is an important discussion to have.

We have also added a sensitivity analysis on the use of a Ramsey discounting approach as recommended by the National Academies for use in Social Cost of Carbon calculations (see response to William Collins) as that is an important alternate approach to discounting.

Minor comments:

1.26: Key criticisms also include the reliance of GWP value on a specified time horizon (that is a value judgement) (e.g. Ocko et al. 2017) and that emissions are not continuous (Alvarez et al. 2012). Would also include citations for each point of criticism that you mention.
http://www.pnas.org/content/109/17/6435

We propose to modify this paragraph as follows:

> Key criticisms of the metric are wide ranging. Some examples include: that radiative forcing as a measure of impact is not as relevant as temperature or damages (Shine et al. 2005); that the assumption of constant future GHG concentrations is unrealistic (Wuebbles et al. 1995, Reisinger et al. 2011); that discounting is preferred to a constant time period of integration (TBD); disagreements regarding the choice of time horizon in the absence of discounting (Ocko et al. 2017);  that dynamic approaches would lead to a more optimal resource allocation over time (Manne and Richels, 2001); that the GWP

does not account for non-climatic effects such as carbon fertilization (TBD); and that pulses of emissions are less relevant than streams of emissions (Alvarez et al. 2012). .

2.1: Definitely one of the reasons, stronger than "likely."

We will delete "likely".

2.3: Please explain upfront *why* you assess the choice of time horizon – as it wasn't even listed in your list of criticisms other than in reference to discounting (and it is problematic aside from discounting as well).

We propose starting the 3rd paragraph with the following sentence:

In this paper, we focus on the choice of time horizon in the GWP as a key choice that can reflect decision-maker values, but where additional clarity regarding the implications of the time horizon could be useful. We also investigate the extent to which the choice of time horizon can incorporate many of the complexities of assessing impacts described in the previous paragraph.

2.3: 100 year was also selected as middle ground from IPCC FAR as values for 20, 100, and 500 years were given.

We suggest modifying the sentence as follows:

The 100-year time horizon of the GWP ($GWP_{100}$) is the has been time horizon most commonly used in many venues, for example in trading regimes such as under the Kyoto Protocol, perhaps in part because it was the middle value of the three time horizons (20, 100, and 500 years) analyzed in the IPCC First Assessment Report.

2.8: Not sure why the word "therefore" is here. A description of why 100 year was selected does not in itself provide justification for why scientists are promoting 20 years. It is because 100 years does not adequately capture near-term impacts as it masks the importance of short-lived climate pollutants in the near-term. There needs to be a better transition from the 100 year discussion to the 20 year discussion.

We will replace "therefore" with "recently".

2.10: Papers such as Ocko et al. 2017 are not pushing for shortened time horizon, they are pushing for a two-valued GWP metric that includes BOTH 20 and 100 year time horizons. Very important distinction that needs to be clarified, as there are efforts (some livestock groups) that push for short time horizon only.

As noted above, we have attempted to present a more nuanced summary.

2.13: Part of the reason that other timescales are not suggested is because of the climate policy community's familiarity with 20 and 100 years. Just as they don't want to adopt a whole new metric, it is

very plausible that they will reject a new time horizon. Since 20 and 100 years are adequate for near- and long-term, pushing for say 30 and 200 year time horizons may be counter-productive.

We propose to clarify that we want more quantitative justifications of timescales in general (whether 20, 100, or 500, or anywhere in between (though also we are including more discussion of some papers such as Boucher 2012 where some analysis along these lines has already been done)

We argue that more focus on quantitative justifications timescales within the GWP structure would be of value, as opposed to qualitative justifications such as a need for urgency to avoid tipping points as in Howarth et al. (2012).

2:20: There are more recent papers that need to be cited that look at the intersection of climate metrics and economics (Tol et al. 2012; Mallapragada and Mignone 2017).
http://iopscience.iop.org/article/10.1088/1748-9326/7/4/044006/meta
http://iopscience.iop.org/article/10.1088/1748-9326/aa7397

See above for our proposed language including some of these references and adding better context, and also our responses to other referees.

2.31: Why CO2 and CH4 only? Justification needed, such as represent the largest long-lived and short-lived climate pollutant contributor's to today's radiative forcing.

We propose adding the following sentence:

The paper focuses on $CO_2$ and $CH_4$ as the two most important historical anthropogenic contributors to current warming, but the methodology is applicable to emissions of other gases and sensitivity analyses consider $N_2O$ and some fluorinated gases.

3.5: Why is a pulse of 28.3 Mt of CH4 used, just bc of 10ppb? Why not today's annual emissions of methane from human activities (around 300-400 Mt)?

See our reply to William Collins for a description of the sensitivity analysis we performed in response to this comment, showing that sensitivity to the size of the pulse is small compared to other uncertainties.

3.12: What radiative efficiencies are used? Should specify this since you go into so much detail of other parameter values. I'm assuming radiative efficiencies are from IPCC AR5 but as you cite in your references, there are more recent calculations in Etminan et al. 2016.

The source of the radiative calculations is described here:

The perturbation of radiative forcing from additional GHG concentrations are based on the equations in Table 8.SM.1 from IPCC AR5. $CH_4$ forcing is adjusted by a factor of 1.65 to account for effects on tropospheric ozone and stratospheric water vapor, as is standard in GWP calculations. $N_2O$ forcing is adjusted by a factor of 0.928 to account for

N$_2$O's impacts on CH$_4$ concentrations, as is also standard in GWP calculations. Baseline radiative forcing is derived from the RCP scenario database.

As noted, Etminan has been cited as an example of updated information on radiative efficiency. Referee 3 also cited Modak et al. as showing the efficacy of methane forcing being lower than for CO$_2$ forcing.

See our reply to William Collins for a description of the sensitivity analysis we performed in response to this comment, showing that sensitivity to even a doubling of methane's radiative forcing would be very small compared to other uncertainties.

3:28: What damages are included by using this function?

This is meant as a simple approximation of all climate damages (sea level, health, ecoystems, etc.).

3:30: Please include citations for the first alternative.

We can cite the National Academies Social Cost of Carbon assessment here.

4.2: Suggest mentioning how these results fit in with scientific literature that has looked at these tradeoffs for decades.

We're not sure what tradeoffs the referee is referring to here: we discuss the Nordhaus GDP growth rates in the context of Gillingham et al.

---

## Author Comment (AC4) · 19 Apr 2018

The manuscript makes a useful contribution to the literature by exploring explicitly how different time scales for GWP relate to GHG equivalence ratios based on damage costs and different discount rates. It is clearly written and highly readable. I have no fundamental concern with the technical approach and quantitative results, but I feel the manuscript needs to work a bit harder to develop its value proposition, discussion of results including sensitivity analysis, and finally the conclusions, before it is fit for publication.

We would like to thank the referee for their comments. For responses comment by comment, see below. All author replies are in red. There is also a summary of new sensitivity analyses that is included at the end of the reply to William Collins.

I'm comfortable with and largely endorse the comments already posted by Bill Collins and anonymous referee #2, and will try not to repeat the specific points they made.

My main concerns where I feel the manuscript needs to work harder are as follows:

1) value proposition: it is mainly in the SI that the authors acknowledge prior work that linked GHG equivalencies based on damage costs and discount rates to GWP. I believe this needs to be brought into the main paper up-front, and the authors need to do a better job explaining where their study adds value to those existing studies. For example, one could argue that their approach is simply a reverse reading of Boucher (2012). I don't think that accusation would be justified, but neither is it justifiable for the manuscript not to recognise the fact that a range of studies have already found that discount rates around 2-3% give the same GHG equivalence between CH4 and CO2 as GWP100. In this context, in the discussion, I would have liked to see a better explanation why their GWP100-equivalent discount rate of 3.3% is higher than that derived by both Boucher and Fuglestvedt et al.

We will bring the SI discussion into the main text, particularly with the comparison to Boucher & Fuglestvedt. See our response to Collins. Better understanding the differences between Boucher & Fuglestvedt is still in progress.

2) discussion of results including sensitivity analysis: in my view, the authors should include an explicit simulation of results if climate-carbon cycle feedbacks following a pulse emission of CH4 are included. The IPCC AR5 and subsequent studies demonstrated that including this results in a significant increase in the GWP100. This is flagged (p7 of the manuscript) but appears not to have been included in the actual sensitivity analysis. It should be fairly easy to modify the radiative forcing calculations to simulate climate-carbon cycle feedbacks and it doesn't have to change the study design at all. There really is no good justification in my view not to include this, other than this is not how the GWP has been defined historically – but from a scientific consistency perspective, it makes no sense to include an effect for one gas (CO2) but not for the other. Including this in the sensitivity analysis (perhaps as a special case, since this is a binary choice rather than something that can be expressed via a pdf) would at least tell us how important this is when we are concerned about choosing GHG equivalencies based on damage functions and discount rates. I could even live with the authors running this only for a central estimate for all other parameters so we can get an order-of-magnitude sense.

See our description of sensitivity analyses at the end of the response to William Collins for a description of the sensitivity analysis we performed in response to this comment, showing that sensitivity to exclusion of the climate-carbon feedback from $CO_2$ had only a small effect. (as well as discussing why we

chose that approach rather than inclusion of the climate-carbon feedback in the $CH_4$ effect). We agree that this was an important analysis to do. The effect of the exclusion was small, due to cancellation when the GWP and the damage ratio are both calculated using consistent assumptions about gas lifetimes and radiative efficiencies.

Related to this, but more difficult to do (hence I would not insist that this is done quantitatively) is consideration of the rate of change as a source of damages. Again this could be parameterised and quantified, but there is a large degree of arbitrariness how much weight to place on rate of change vs amount of change. The manuscript would be much stronger though if it could demonstrate under what circumstances including the rate of change might affect the conclusions, or whether the conclusions might be robust even if rate of change damages have been incorporated within reasonable bounds.

We have implemented a crude rate of change analysis within our framework (see discussion in the sensitivity analysis at the end of the reply to William Collins for details), and determined that under RCP6 inclusion of even extreme damage estimates due to rate of change have little effect on implicit timescales. However, under the RCP3PD scenario, the incorporation of rate of change has a larger effect, reducing the implicit timescale by almost half under the rate of change damage parameters that may be more realistic in magnitude. Our approach is not sophisticated enough to become a major component of the paper, but the results would be worth noting in a sentence or two, along with reference to Bowerman (2013) which provides a good explanation for the reasons why we see this result.

3) interpretation and conclusions: I would endorse some of the comments made by anonymous reviewer #2, that the authors are effectively beating up a strawman. Yes some people have argued that we should simply 'switch' to GWP20, but the more intelligent arguments are all for considering the effect of multiple alternative time horizons to inform abatement decisions and policy choices. See e.g. the conclusions in Levasseur et al 2016 (doi: 10.1016/j.ecolind.2016.06.049) regarding the use of multiple time horizons and metrics in lifecycle assessment. The discussion and conclusions need to add quite a bit of nuance to reflect what those studies actually say, and hence the degree to which this manuscript challenges their conclusions or simply adds another dimension that can help choosing the right metric for the right purpose.

We have added more nuance to our description (see responses to other referees), as well as additional context to our results.

There are two additional points that the discussion and conclusion needs to address:

(a) one is that a key context in which GHG metrics are used are in emission trading schemes, and to help governments evaluate policy choices that directly affect near- term commercial decisions, i.e. policy that would "alter the use of capital in the private sector". So there are very different contexts in which GHG metrics are actually used in climate policy and where different discount rates are commonly applied, and the paper would be stronger and more relevant if it recognised and addressed these explicitly.

We recognize that there is common justification (e.g., OMB Circular A-4) for the use of one discount rate for social problems and another for policies that "alter the use of capital in the private sector". We would argue, however, that comparing the relative impacts of $CO_2$ v. $CH_4$ should always been considered a social problem in this context, regardless of whether it is being used in decisions regarding capital or by governments within emission trading programs.

We will consider how to address this within the paper, but it is a somewhat tricky topic, as it can pose consistency challenges that go well beyond the implications of this paper. For example, a decisionmaker deciding between investing in a CNG vehicle and a gasoline vehicle might want to look at vehicle costs, maintenance, and fuel prices under a high discount rate to reflect the opportunity cost of that investment, but, because the benefits of the GHG abatement are not received by the decisionmaker but by society as a whole, it could be argued that the latter issue should be considered with a societal discount rate. It is unclear how to bring those two monetization streams into a single analysis given the difference in discount rates.

(b) The second is a recognition that IAMs used to design cost-minimising emission pathways often use a discount rate of 5%. Given that a(nother) key use of GHG metrics is to help IAMs make trade-offs between different gases with different mitigation costs, this should enter into the discussion in this paper. I don't think this materially changes the conclusions since we know that different GHG metrics don't have a massive effect on total mitigation costs (although there is a systematic effect especially when moving towards GWP20), but the issue is not trivial especially for countries or sectors with non-negligible non-CO2/SLCF sources. Some discussion on this is needed.

We plan to add more discussion regarding some of the IAM-based tradeoff work such as van den Berg et al. (2015), Reisinger et al. (2013), and Smith et al. (2013). The common use of a 5% discount rate might relevant to that discussion.

I believe that all the above points (with the exception of quantifying the effect of including climate-carbon cycle feedbacks for CH4) can be addressed by a careful revision of the text itself. The manuscript needs to avoid what currently appears as the too- simplistic conclusion that "actually, GWP100 is largely ok, let's move on" (which is how I read P8L22). The fundamental finding from virtually all metrics papers is that the right metric depends on the application, and hence it is rather jarring to read a conclusion that continued use of GWP100 is 'reasonable' without any caveat.

We will endeavor to add more nuance to the conclusion, though the authors do believe that this analysis is fairly strong support for a 100 year GWP in many contexts, given acceptance of a few key assumptions. These key assumptions can be made more clear, and might be as follows (but with some improvement in phrasing):

      1) The context of comparing pulses of emissions. This may not hold true within a framework of, for example, an absolute temperature target or other global decision-making process.

      2) A valuing of time similar to a 3% discount rate

      3) Given the assumptions about damage functions, future scenarios, and other parameters made within this paper.

      4) Not considering additional impacts such as CH4-O3 or acidification (which could be addressed with post-hoc adjustments of relative weights)

5) Not considering how the metric functions within the context of the broader economic system (which could be better addressed within an IAM)

We do recognize that there are other uses of metrics such as the AGTP to produce a mechanism by which future temperatures changes resulting from emissions pulses can be quickly estimated, but that is a separate issue from the relative impacts addressed by this paper.

I am not repeating the above points in my specific comments below and would be happy for the authors to decide how they can best address them.

Specific comments:

P1L22: insert 'emission' after gases – we're talking about emission metrics here

Will do.

P2L3: 'endorsed' is too strong in my view for the UNFCCC – 'used' is more factual, I cannot recall an explicit endorsement in the sense that the UNFCCC would have explained and justified its choice.

We propose language such as:

The 100-year time horizon of the GWP ($GWP_{100}$) is the has been time horizon most commonly used in many venues, for example in trading regimes such as under the Kyoto Protocol, perhaps in part because it was the middle value of the three time horizons (20, 100, and 500 years) analyzed in the IPCC First Assessment Report.

P2L11: I believe the correct term for GTP is Global Temperature CHANGE Potential

In our defense, AR4 and other sources also refer to the GTP as the Global Temperature Potential: e.g., "2.10.4.2 The Global Temperature Potential"). But "change" seems to be more standard, including in AR5, and so the updated manuscript will reflect that.

P2L12: the reason why GTP downplays SLCFs is not primarily that it is temperature based but that it is a point metric. iGTP is very similar to GWP.

We are updating language to better differentiate integrated versus endpoint metrics. We note, however, that Brazil and New Zealand specifically suggest the GTP, not the iGTP, and justify it based on the temperature argument.

P2L22-27: editorial only: I prefer if introductions don't include the conclusions but rather focus on making the point of why the conclusions are worth having.

We will edit accordingly.

P3L15: shouldn't the N2O effect on CH4 forcing depend on the RCP pathway? Perhaps this was done but this isn't clear to me from the text.

The particular adjustment for $N_2O$ cited here is based on 8.SM.11.3.3, which estimates that emissions of 100 molecules of $N_2O$ will lead to a destruction of 36 molecules of CH4. For this effect, the RCP pathway does not matter.

This is in contrast to the overlap between $N_2O$ and $CH_4$ radiative absorption bands, which does depend on the RCP pathway, and is captured in the radiative forcing equations used for this paper (and for the GWP).

P4L9: 'future years are cooler than present': helpful if you could indicate what years we are talking about (presumably you mean after 2200 or thereabouts, depending on the reference period/warming – meaning that much of those will be heavily discounted anyway).

Under RCP3PD, with climate sensitivity of 3.92, and a forcing imbalance of 0.84, temperatures drop below the starting temperature only 458 years into the analysis. Therefore, in this case, when the damage function is relying on temperature change since present (rather than since 1950-1980 or earlier), these years would need to be set to zero. It is possible that under different climate sensitivities or forcing imbalances, this could occur somewhat earlier, but we believe that the referee's intuition that this effect will be negligible due to discounting is correct.

P4L21: here and later, please clarify where you truncate your damage calculations (when I read this sentence, I thought you truncate at 2300, but later (P5L14) it seems you truncate at 2500). You note below that this may matter for very low discount rates. Can you quantify/illustrate this?

We will clarify that the graphs are to 2300, but the calculations do go to 2500. We also include text regarding the size of the effect as follows:

> Even at a 2% discount rate, 95% of the $CO_2$ damages come in the first 287 years. At discount rates lower than 2%, however, truncation effects can account for errors in damage ratio estimates of greater than a percent, indicating that longer calculation timeframes may be necessary to capture the full effect of the emissions pulse.

P6L4: I think the entire sensitivity discussion should note that projecting damages multiple centuries into the future is increasingly fraught with difficulties. The AR5 chose not to evaluate GWP500 because the authors felt that (deep) uncertainties were simply too large – but here you evaluate damages from temperature responses from forcing 500 years into the future? At least a comment on this is needed here – the discussion of what percentage of total damages occurs up to a given year for CH4 and CO2 is useful in this context and could be linked to this point about uncertainty.

See the response above regarding the low percentage of damages that occur after the first 290 years even under a 2% discount rate. But we can also include a note about the challenges of projecting damages many decades into the future. In particular, the GDP projections are a key component of the analysis, and therefore the dramatic uncertainties about future economic growth are particularly important. We also propose to include a sentence noting that uncertainties as shown in Figure 2 are larger at low discount rates in part for this reason.

P8L2: I feel the statement "We note no metric designed to tradeoff emission pulses is consistent with stabilization" is too strong. Of course, no metric in itself delivers stabilisation, but almost any metric can be used wisely enough to help countries achieve stabilisation.

We propose to move some of the following discussion into the main text to make a more in-depth discussion of this particular point, hopefully in a more nuanced fashion. We particularly wanted to highlight the challenge of using, for example, a GTPX metric to achieve a cost-optimal approach to achieving a temperature target in a given year, but which when used in that way would lead to increasing challenges of maintaining temperatures at that target in future years (in part because the short-term GTP would lead to more SLCF abatement relative to $CO_2$ abatement than would be indicated by, for example, a GWP100). This is in addition to the fact that any SCLF/$CO_2$ trading using whatever pulse-based metric post-stabilization will lead to moving away from stabilization. Allen et al. 2016 has a particularly elegant approach to this question. Though all of this is perhaps not central to the main analysis in the paper...

A number of authors have recognized that the GWP is not designed to achieve stabilization goals (Sarofim et al. 2005, Smith et al. 2012, Allen et al. 2016). Some actors (Brazil INDC, 2015) have claimed that certain metrics such as the Global Temperature Potential (Shine et al. 2005) or the Climate Tipping Potential (Jorgensen et al. 2014) are more compatible with a stabilization target such as 2 degrees C because they are temperature based. However, these metrics are also not designed to achieve stabilization goals, but rather to achieve a temperature target in a single given year. The challenge is that in any year after stabilization, any trading between emission pulses of carbon dioxide and a shorter-lived gas will cause a deviation from stabilization. For example, trading a reduction in methane emissions for a pulse of $CO_2$ emissions will lead to a near term decrease in temperature, but also a long-term increase in temperature above the original stabilization level.

One solution to the problem is a physically-based one. Allen et al. (2016) suggest trading an emission pulse of carbon dioxide against a sustained change in the emissions of a short-lived climate forcer. This resolves the issue of trading off what is effectively a permanent temperature change against a transient one. However, the challenge becomes one of implementation, as current policy structures are not designed for addressing indefinite sustained mitigation. Alternatively, a number of researchers (Daniel 2012, Jackson 2009, Smith et al. 2012) suggest addressing $CO_2$ mitigation separately from short-lived gases. Such a separation recognizes the value of the cumulative carbon concept in setting GHG mitigation policy (Zickfeld et al. 2009).

---

## Author Response (AR1)

**Reviewer 1 Response:**

This paper usefully links the well-used GWP metric with more economic comparisons of the ratios of damages. It is suitable for publication and I only have minor comments.

We would like to thank the referee for their comments. For responses comment by comment, see below. All author replies are in red. There is also a summary of new sensitivity analyses that is included at the end of this comment reply.

The relationship to the Paris goals are only briefly alluded to. I suggest including a longer discussion of the differences between: a temperature limit, a long-term temperature goal, and least economic cost. Presumably neither a temperature limit nor a long-term temperature goal are optimal economically using the damage function here? Is this a problem with the Paris agreement? The choice of metric depends entirely on the choice of target, and the authors here are implicitly assuming that least economic damage is the most important target. The authors dismiss short-term GTPs as implying unrealistic discount rates, which of course they do for an economic damage target. However, if the Paris agreement is taken to imply that temperatures should not be allowed to exceed 2 degrees, then a GTP with a time horizon ending at the time of peak warming (20-30 years) is an entirely appropriate metric. Similarly, if the Paris agreement is taken to mean a long-term goal to stabilise at 2 degrees then GWP* is the appropriate metric.

The reviewer brings up an interesting question regarding the goals of global climate agreements. To some extent, the optimal design of a global goal is beyond the scope of this manuscript, but given that one important use of climate metrics is within cap and trade programs that are meant to help achieve these global goals, some discussion would be appropriate. To that end, we brought some of the text that was originally in the SI into the main manuscript, with some updating and editing:

> While this paper focuses on a cost-benefit approach, there is also a potential need for cost-efficiency approaches, particularly in regard to stabilization targets such as 2 °C. However, a number of authors have argued that pulse-based metrics such as the GWP are not well-suited to achieve stabilization goals (Sarofim et al. 2005, Smith et al. 2012, Allen et al. 2016). Some actors (Brazil INDC, 2015) have claimed that certain metrics such as the Global Temperature Potential (Shine et al. 2005) or the Climate Tipping Potential (Jorgensen et al. 2014) are more compatible with a stabilization target such as 2 °C because they are temperature based. However, any pulse-based approach faces at least two major challenges related to stabilization scenarios. The first is that as a temperature target is approached, a dynamic approach will shift from favouring long-lived gas mitigation to favouring short-lived gases. While this shift may be optimal for meeting a target in a single year, it will be sub-optimal for any year after that year. The second challenge is that once stabilization has been achieved, any trading between emission pulses of carbon dioxide and a shorter-lived gas will cause a deviation from stabilization. For example, trading a reduction in methane emissions for a pulse of CO2 emissions will lead to a near term decrease in temperature, but also a long-term increase in temperature above the original stabilization level. One solution to the problem is a physically-based one. Allen et al. (2016) suggest trading an emission pulse of carbon dioxide against a sustained change in the emissions of a short-lived climate forcer. This resolves the issue of trading off what is effectively a permanent temperature change against a transient one. However, the challenge becomes one of implementation, as current policy structures are not designed for addressing indefinite sustained mitigation. A second solution is a dynamically updating global cost potential approach that optimizes shadow prices of different gases given a stabilization constraint (Tol et al. 2016), but again, implementation would be challenging. Alternatively, a number of researchers (Daniel et al. 2012, Jackson 2009, Smith et al. 2012) suggest addressing CO2 mitigation separately from short-lived gases. Such a separation recognizes the value of the cumulative carbon concept in setting GHG mitigation policy (Zickfeld et al. 2009). However, this approach requires a central decisionmaker and loses the

"what" flexibility that makes the use of metrics appealing (Bohringer et al. 2006). In economic terms, a temperature based target is equivalent to the assumption of infinite damage beyond that threshold temperature, and zero damages below that threshold (Tol et al. 2016).

The overall formula for the damage function needs to be shown as a function of temperature, discount rate, GDP etc.

The following text was added:

Damages as a percent of GDP were calculated by multiplying a constant times by the square of the temperature change since the baseline period squared. E.g., $D(2050) = a*\Delta T(2050)^2*GDP$. The net present value is then calculated using the discount rate such that $NPV(D(t)) = D(t)/(1+r)t-2010$.

Page 1, line 26: Maybe a different word other than "endpoint" could be used so as to avoid confusion with the later discussion of integrated and endpoint metrics.

We replaced endpoint here with "measure of impact"

Page 2, first paragraph: The main difference between GTP and GWP is the difference between endpoint and integrated metrics. This should be brought out more in this paragraph. The iGTP could be mentioned as it is more similar to GWP than GTP.

Integrated metrics are addressed in the following:

Mallapragada and Mignone (2017) present a similar framework and also note that metrics can consider a single pulse of a stream of pulses over multiple years. Several authors have recognized that under certain simplifying assumptions, the GWP is equivalent to the integrated GTP, and therefore any timescale arguments that apply to analyses of one metric would also apply to the other (Shine et al. 2005, Sarofim 2012).

Page 2, second paragraph: Boucher ESD 2012 should also be discussed for economically-based equivalences.

Boucher 2012 and Fuglesvedt 2003 are now discussed in more depth in the introduction, sensitivity analyses, and conclusion sections.

Page 4, line 1. These GDP pathways should be shown (maybe in the supplement).

We are including a graph of the GDP pathways in the SI.

Page 4, line 6. It is not obvious why 1951-80 should be chosen as a baseline. A problem with damage functions that are non-linear functions of temperature is that a point needs to be chosen when temperatures were optimal.

We agree that there is no clear best baseline: that is why we did a sensitivity analysis using a baseline of 0 (effectively, that damages are a function of temperature change relative to 2010) and a baseline of 0.8 degrees (assuming that damages are a function of temperature change since preindustrial times) were used. Table 1 indicates that this choice can have a difference of up to 26% in terms of damages.

Figure 1: I was surprised by the shape of 1 (c). Why does the damage from CH4 keep increasing? Is the damage an integral quantity, or is this increase purely due to an exponential increase in GDP? In 1(d) the damage decreases. Is this because the

discount rate is larger than the GDP growth? With the GDP growth of 2.06% would a discount rate of less than 2% give an increasing damage for a gas like CO2 with a non-decaying component?

The following was added to address this question:

In the case of $CH_4$, damage peaks in 2032 and declines until 2080 as a result of the short lifetime of the gas. The increase in damages after 2080 is due to the component of the temperature response function that includes a 409 year timescale decay rate, such that after 100 years the decrease in the $\Delta T^2$ component of the damage equation is about 0.5%/year, and because that decay rate is slower than the rate of GDP growth, net damages grow.

Page 6, line 13: I don't think "exponential function of temperature" is the right term for temperature raised to a power.

Changed to "polynomial function".

Table 2: The ranges (either 25%-75% or 10%-90%) need to be shown as well as the central value. These are quite large for the timescales and may well include 1.0 for many of the damage ratios.

We have updated the table as follows, including uncertainty ranges for $CH_4$: Uncertainty ranges for the longer-lived gases are more challenging, as the non-monotonicity adds complications.

**Table 1: Parameter sensitivity analysis: Examining the sensitivity of the GWP-discount rate equivalency as shown in the uncertainty ranges in Fig. 2 as a function of the individual parameters of the calculation. The ratio is calculated as the ratio of the median of the estimated GWPs given the highest and lowest value of each parameter. Results in this table are derived assuming as discount rate of 3%.**

| Gas | Lifetime | Optimal timescale | $GWP_{100}$/damage ratio | $GWP_{20}$/damage ratio |
|---|---|---|---|---|
| $CH_4$ | 12.4 | 120 (84-172) | 1.15 (1.52-0.87) | 3.4 (4.49-2.57) |
| $N_2O$ | 121 | *52 | 0.85 | 0.84 |
| HFC-134a | 13.4 | 115 | 1.11 | 3.2 |
| HFC-23 | 222 | *105 | 0.71 | 0.62 |
| PFC-14 | 50000 | >400 | 0.62 | 0.45 |

**Table 2: Optimal timescale of non-$CO_2$ gases. Implicit timescale evaluated for non-$CO_2$ gases with the GWP to damage ratio for the two most common GWP timescales. Asterisks indicate no exact match between GWP ratio and damage ratio, closest value is given instead. The third and fourth columns show the ratio of the GWP for a given gas to the calculated damage ratio. Interquartile uncertainty ranges are presented for the timescale and damage ratios for $CH_4$. Results in this table are derived assuming as discount rate of 3%.**

Page 6, line 25: GWP100 seems to agree very well with the 3% discount rate within the uncertainty rather than overvaluing or undervaluing.

We have modified the text "However, the analysis shows that the 100-year timescale is consistent, within the interquartile range, with the 3% discount rate that is commonly used for climate change analysis."

Page 6, line 25-29: I didn't understand this sentence. Are you saying that the uncertainty in GWP100 is such that it covers agreement with the 3% discount? If so, that seems to contradict the previous sentence which suggested a under/overvaluing

5   We chose to delete the sentence in question. Upon reflection, the uncertainty in the GWP100 is due to factors like uncertainty in radiative efficiency, but (as shown in a sensitivity calculation) the timescale is not sensitive to radiative efficiency uncertainty.

Additional Sensitivity Analyses in response to various Referee Comments:

The four referees raised a number of points, several of which were inter-related. In response to these points, we have performed a number of additional sensitivity analyses, which will be detailed here. One lesson from these analyses: the analysis is robust to a surprising number of changes. This relates to the fact that any change in the analysis which changes both the GWP and the damage ratio generally cancels out in terms of calculating the relationship between discount rate and timescale: e.g., most factors in the causal chain from emissions through radiative forcing. That includes the size of the emissions pulse, the radiative efficiency of the gas, and the lifetime of the gas (within certain limits). Note that changes in these parameters may well change the best estimate of the relative importance of reducing methane compared to $CO_2$, but they do not change the implicit timescale that is the focus of this paper.

In contrast, anything which changes only the damage ratio can have a larger impact on the implicit timescale. This includes the factors which have the largest influence on the uncertainty, as reflected in Table 1: the rate of GDP growth, the damage exponent, the scenario (because the GWP assumes constant concentrations), the baseline temperature from which damages are calculated, and the climate sensitivity.

A third category is influences which are specific to a given gas, whether the efficacy of that gas (e.g., Modak et al. noted by Referee #3), or the health effects of methane-related ozone productions. Like changes to radiative efficiency or lifetime, these influences can also change the best estimate of the relative importance of reducing methane compared to $CO_2$, but because they are specific to a single substance and not generalizable between substances, they would not be appropriate for calculating implicit timescales.

Herein, we summarize the additional sensitivity analyses:

Pulse Size:

In response to a comment by Referee #2, we performed a sensitivity analysis on the size of the emissions pulse. Using 373 MMT (one year's emissions according to Saunois 2016, http://iopscience.iop.org/article/10.1088/1748-9326/11/12/120207, where global emissions are 559 MT of which 2/3rds are anthropogenic), the relative damage ratios of $CH_4/CO_2$ are the following:

| quartile | 25% | 50% | 75% |
|---|---|---|---|
| Damage Ratio | 18.80 | 25.05 | 32.69 |

A similar analysis for the 28.3 MMT used in the paper, performed in response to a comment by reviewer 1, yields

| quartile | 25% | 50% | 75% |
|---|---|---|---|
| Damage Ratio | 18.84 | 25.12 | 32.86 |

The differences between the two analyses are less than 1%. We will note this sensitivity analysis in the paper. Arguments can be made for either the larger quantity or the smaller quantity depending on the purpose of the analysis. When GWPs are used to inform decision making, these decisions are often regarding mitigation actions at the national or sub-national scale, which is closer to the 28 MMT scale than the 370 MMT scale. When GWPs are used to compare $CO_2$equivalents for global scenarios, then the global annual number might be more appropriate. But in any case, this appears to be an uncertainty much smaller than any of the others considered in this paper.

Radiative Efficiency:

Referee #2 also raised a question regarding the use of the IPCC AR5 radiative efficiencies rather than the more recent Etminan et al. 2016 article. In order to test the general sensitivity of this approach to changes in the radiative efficiency of methane, a sensitivity analysis was performed, wherein forcing from $CH_4$ was doubled compared to the standard calculation. Relative damages of $CH_4$ and $CO_2$, as might be expected, also doubled:

| quartile | 25% | 50% | 75% |
|---|---|---|---|
| Damage Ratio | 37.98 | 50.35 | 65.50 |

However, calculated GWPs based on the updated radiative forcing also double: the new GWP100 being 56.79, and the new GWP20 being 166.92. The optimal timescale using the central parameters ends up being 119.79 years, under 1/10th of a percent different than the original calculated timescale of 119.85 years.

We can also mention this in the paper.

Consistent Climate Carbon Feedbacks:

Referee #4 points out, with good cause, that it would be a good sensitivity analysis to calculate an implicit timescale using consistent assumptions about climate-carbon feedbacks for both carbon and methane lifetimes (whereas, in the main body of the paper, we used a $CO_2$ lifetime that included climate-carbon feedbacks, and a methane lifetime that did not, consistent with the approach of the IPCC AR4 and the GWP values reported in the main body of chapter 8 in IPCC AR5). Using the "no-climate-carbon-feedback" $CO_2$ equation from Gasser et al. 2017, we can calculate $CH_4/CO_2$ damage ratios:

| quartile | 25% | 50% | 75% |
|---|---|---|---|
| Damage Ratio | 20.57 | 27.16 | 35.23 |

This can be compared to the original damage ratios from the paper:

| quartile | 25% | 50% | 75% |
|---|---|---|---|
| Damage Ratio | 18.84 | 25.12 | 32.86 |

The damage ratios are about 8% larger under the consistent no-CC-feedback case than in the $CO_2$-feedback/$CH_4$-nofeedback case, due to what is now a shorter $CO_2$ lifetime. But a GWP calculated with consistent assumptions regarding climate-carbon
5   feedbacks also changes. The no-feedback GWP100 is now 30.74 rather than 28.64, a 7% increase. These two effects largely cancel out, leaving the effect on implicit timescales smaller, again, than most of the other sensitivities examined in the paper.

| quartile | 25% | 50% | 75% |
|---|---|---|---|
| consistent no cc feedback timescales | 170.56 | 117.9 | 83.38 |
| original analysis | 170.87 | 118.30 | 83.81 |

So while the damage-ratio differences are on the order of 8%, the implicit timescale differences are less than 1%. Similar to the impact of changing methane's radiative forcing, a large impact on damage ratios has a small impact on implicit timescales because the update happens in both the calculation of the damage-ratio AND in the calculation of the GWP, and
10   therefore cancels out to a large extent.

We show here a calculation that was consistent in not including climate carbon feedbacks in either damage calculation. While it would be possible to do a similar consistent calculation with climate carbon feedbacks in both damage calculations, implementation of the climate-carbon feedback for methane is not without challenges. Such a calculation requires first calculating of the temperature impact resulting from an emissions pulse over the full time period (as done in the original),
15   and then, for the temperature change in each year, requires a calculation of the carbon perturbation resulting from that temperature change over the remainder of the time period, and then calculating the additional forcing and temperature change resulting from that carbon perturbation. Therefore, we decline to do this calculation at this time (absent an already available methane lifetime equation with the CC feedback built in, which the authors do not have on hand). The authors are already on record as preferring a GWP metric that does not include CC feedbacks in either the $CO_2$ or the $CH_4$ response
20   rather than inclusion of CC feedbacks in both $CO_2$ and $CH_4$ responses – see the response by Sarofim, Giordano, and Crimmins to the Gasser paper for a more detailed explanation (https://www.earth-syst-dynam-discuss.net/esd-2016-55/esd-2016-55-SC1.pdf)

Ramsey Discounting:

25   Of all the sensitivity analyses (other than discount rate), the calculation of implicit timescales was most sensitive to changes in the assumption about future GDP growth rates. The original draft raised the question as to what extent this sensitivity to GDP growth might diminish were the discount rate to be a function of GDP, as in a Ramsey discounting framework. In this case, a high GDP growth rate (which implies large future damages, and therefore a low methane/CO2 damage ratio) would be counteracted by a high discount rate which would be expected to lead to a high methane/CO2 damage ratio.

For this sensitivity analysis, the Ramsey discounting approach was calibrated to yield an average discount rate for the first 30 years of the analysis of 5% for the reference GDP growth rate, with a pure rate of time preference of 0.01% (a very low pure rate of time preference being consistent with the assumption that, holding consumption constant, all generations should be given equal weight). That required an elasticity of marginal utility of consumption of 1.53.

Interestingly, the ratio of the median damage estimate from the low GDP scenario to the high GDP scenario was still about a factor of 2, except that now the low GDP scenario leads to the lowest damage ratio and the high GDP scenario leads to the highest damage ratio: effectively, in the Ramsey case, the discount rate effect overwhelms the GDP growth rate effect. Note that the equivalent timescale in the median GDPref case under this set of assumptions is 135 years.

| GDP scenario | | GDPlow | GDPref | GDPhigh |
|---|---|---|---|---|
| Discount rate, first 30 years-> | | 2.5% | 5% | 7.5% |
| Discount rate, full run-> | | 0.8% | 1.5% | 2.3% |
| Ramsey discounting approach | 25% | 11.28 | 15.71 | 21.78 |
| | 50% | 16.08 | 22.72 | 31.05 |
| | 75% | 18.42 | 26.40 | 35.16 |
| | | 3% | 3% | 3% |
| Original constant discount rate approach | 25% | 29.96 | 21.11 | 14.46 |
| | 50% | 38.69 | 27.58 | 18.70 |
| | 75% | 42.52 | 30.29 | 19.81 |

Rate of Change Calculation:

Referee #4 also brought up the idea of adding in damages from rate of change of temperature as well as from absolute temperature change. This was an important addition in, for example, Manne & Richels (2001) where it turned a dynamic optimization metric from one that didn't value methane until a decade or two before threshold temperatures were reached to a metric which had fairly constant value over time.

In order to test this, we added code that included damages from rate of change. The damage is calculated by taking the square of the rate of change and multiplying by the GDP and a constant – similar to the damage calculation for absolute temperature. For the first test, peak rate of change damages under the central scenario were calibrated to be equal to the damages in 50 years (2060), where there is 1 degree of temperature change (above the temperature offset) under central parameter estimates. This led to an increase in damage ratio of 2.4%.

A second test, with peak rate of change damages 10 times as large, led to an increase in the damage ratio of 5.1%. That yields a timescale change from 120 years to 112 years.

The small impact of the rate of change damage is likely due to timing of peak rates of change under RCP6. In this analysis, there is an initial high rate of change for the first few years, followed by a secondary (though smaller) peak about 60 years in. Because of the GDP growth in the interim, damages at the 60 year secondary peak are higher than damages in the initial years. Therefore, reducing the rate of temperature growth 60 years into the run is more important than reducing the rate of growth at the beginning of the run, and for this reason, reducing short-lived forcer emissions does not have large advantages over reductions in $CO_2$ emissions.

However, this raises a question about the importance of the scenario assumption. Therefore, we also tried the same exercise with the RCP3PD scenario as the baseline. In RCP3PD, in this analysis, the rate of temperature change is at its highest in the first years of the scenario, which means that the increase in rate of change due to additional radiative forcing in those early years can have a disproportionately large impact, favoring short-lived climate forcers. Note that because RCP3PD cools in later years, we chose to set the damages resulting from a negative rate of change to zero.

In this case, the first rate of change analysis yields an increase in the damage ratio of 53%, and the second analysis yields a damage ratio increase of 85%. This yields a timescale change from 94 years (the timescale for RCP3PD without rate of change damages) to 54 years to 42 years.

This is consistent with some other literature such as Bowerman et al. (2013), which suggests that under stringent $CO_2$ mitigation scenarios, the rate of change peaks in the near future, and therefore reduction of short-lived climate forcers can be particularly valuable – but that in scenarios where $CO_2$ emissions continue to rise in the near term, the rate of change peaks further in the future, and therefore delay in short lived climate forcer mitigation will lead to the greatest reductions in peak rate of change.

However, this particular sensitivity analysis was rather crude. In order to improve it, we would have to develop a better reasoning for choosing the parameters involved (e.g., the damage exponent for rate of change, and the damage constant), but also investigate how well our approach models near-term rate of change compared to more complex models, as that is of importance for this calculation.

**Reviewer 2 Response:**

The paper provides an interesting analysis on the connection between GWP timescale and discounting rates. However, I think that before publication, some major issues need to be addressed.

We would like to thank the referee for their comments. For responses comment by comment, see below. All author replies are in red. There is also a summary of new sensitivity analyses that is included at the end of the reply to William Collins.

First, while the paper acknowledges a lot of recent articles that discuss GWP, it does not adequately discuss recent articles that look at climate metrics in an economic frame- work (such as Tol et al. 2012 and Mallapragada and Mignone 2017). There needs to be more incorporation of these types of studies to show how this study builds on the existing literature.

We have moved SI material into the main text, and added several references and additional discussion.

Second, the authors seem to misunderstand the messages of several recent studies, such as Shindell et al. 2017 and Ocko et al. 2017. These studies are not advocating for a shorter time horizon for GWP, as this paper implies in both the main text and the supplemental information. Rather, they are advocating for using BOTH short AND long-term time horizons to capture the full scope of climate impacts over all timescales – a key distinction that is not depicted in the text. The paper in its
10   current forms criticizes these studies for something that they are not doing. Further, the authors frame their motivation around the fact that studies are advocating GWP20 to then show that GWP100 fits better with discount rates, but because these studies are not simply advocating GWP20, it makes the authors appear naïve to the existing literature. Further, there is a strong reason behind why other timescales are not promoted which needs to be acknowledged (it is not simply a lack of quantification in research efforts) – that just as it is difficult to move the policy community away from the comfortable
15   GWP, it is reasonable to believe that it will be equally as difficult to move the community away from 20 and 100 year timescales of which they are also most familiar with.

We will add clarification that in some cases authors are often suggesting presentation of short (about 20 years) timescales alongside, not in place of, 100 year timescales. However, we do feel that "promoting more emphasis on shorter time horizons" is an accurate description, as providing both GWP20 and GWP100 (for example) is "more emphasis" on shorter
20   time horizons than just providing GWP100. Presented with values for 2 time horizons, one might also expect a decision-maker to have some probability of choosing to use only the shorter one, or to weigh them equally, which would look like a GWP60 (for comparison, a GWP60 would have an median equivalent discount rate of 5% using our methodology). Meanwhile, some authors (e.g., Howarth) do explicitly state that use of the 20 year GWP would better account for relevant climate impacts than 100 year GWPs, and a number of NGOs have followed suit. We have added a sentence about Ocko et
25   al. to reflect some more nuance:

These studies each have different nuances regarding their recommendations – for example, Ocko et al. (2017) suggest pairing the GWP100 with the GWP20 to reflect both long-term and near-term climate impacts – and therefore there is no simple summary of the policy implications of this body of literature, but it is plausible that more consideration of short-term metrics would result in policy that weights near-term impacts more heavily.

30   Third, it would be great if the damages function description went into more details about what is included in "damages." For example, I believe the authors make it clear later on that health or agriculture impacts from methane were not included. So what is included? Those damages are part of what makes near-term impacts so important to reduce, which justifies the use of a shorter time horizon.

Damages in this case are only climate-related damages. Inclusion of damages due to, for example, the health impacts of $O_3$
35   are relevant for policymaking purposes as one of the authors has argued elsewhere (see, e.g., Sarofim, Waldhoff, and Anenberg), but we would argue are not appropriate for the timescale discussion. One support for keeping timescale & non-climate effects separate is that if the timescale is adjusted to increase the value of methane to account for the methane-ozone effects, then it will equally increase the value of HFC-134a which has a similar lifetime: but HFC-134a does not have an equivalent ozone-effect. Therefore, instead, the relative climate effects of gases should be calculated using this timescale
40   approach, and then the value of reducing methane can be increased to account for its ozone effects (and the value of reducing $CO_2$ can be adjusted to account for fertilization and acidification effects).

We have added some more discussion about how to address non-climatic impacts like the methane-ozone effect.

Finally, I wonder about the argument that we should select a time horizon based off of appropriate discount rates. What if the GWP timescale tells us which discount rates are more appropriate? Why is it necessarily the other way around? The literature on appropriate discount rates is vast and its value is debated as much as GWP timescale selection. The paper makes
5 it seem like there is solid agreement on appropriate discount rates but not GWP timescales, but both are subject to similar challenges and debates.

Our analysis can be used in either direction, as is discussed in the paper:

Here we focus on what discount rates are consistent with a GWP time horizon in order to show the discount rates implied by common choices of GWP timescales. The converse calculation is relevant for an audience that has a
10 preferred discount rate and is interested in the implied GWP timescale.

While we agree that there is no single consensus on an appropriate discount rate, we do think that that the framing of discount rates is good way to formally demonstrate the implications of GWP timescales for valuation over time. Part of the impetus of this paper was due to the possibility that some proponents of shorter timescales might not recognize the implicit discount rate embodied in the timescale choice, and might not consider a high discount rate to be desirable. We recognize
15 that the discount rate implications calculated in this paper are a result of many decisions regarding parameters and analytical approaches, and that other approaches might yield different results, but we think that this is an important discussion to have.

We have also added a sensitivity analysis on the use of a Ramsey discounting approach as recommended by the National Academies for use in Social Cost of Carbon calculations (see response to William Collins) as that is an important alternate approach to discounting.

20 Minor comments:

1.26: Key criticisms also include the reliance of GWP value on a specified time horizon (that is a value judgement) (e.g. Ocko et al. 2017) and that emissions are not continuous (Alvarez et al. 2012). Would also include citations for each point of criticism that you mention. http://www.pnas.org/content/109/17/6435

The paragraph has been modified as follows:

25 Criticisms include the following:: from arguments that radiative forcing as an endpoint measure of impact is not as relevant as temperature or damages (Shine et al. 2005); , to critiques of that the assumption of constant future GHG concentrations (Wuebbles et al. 1995, Reisinger et al. 2011) is unrealistic;, to the position that discounting is preferred to a constant time period of integration (Schmalensee, 1993); disagreements about the choice of time horizon in the absence of discounting (Ocko et al. 2017);, to the view that dynamic approaches would lead to a
30 more optimal resource allocation over time (e.g., Manne and & Richels, 2001, Manne & Richels 2006), and to the fact that the GWP does not account for non-climatic effects such as carbon fertilization or ozone produced by methane (Shindell 2015); and that pulses of emissions are less relevant than streams of emissions (Alvarez et al. 2012).

2.1: Definitely one of the reasons, stronger than "likely."

35 "likely" was deleted.

2.3: Please explain upfront *why* you assess the choice of time horizon – as it wasn't even listed in your list of criticisms other than in reference to discounting (and it is problematic aside from discounting as well).

We now start the 3rd paragraph with the following sentence:

> In this paper, we focus on the choice of time horizon in the GWP as a key choice that can reflect decision-maker values, but where additional clarity regarding the implications of the time horizon could be useful. We also investigate the extent to which the choice of time horizon can incorporate assess the choice of time-horizonmany of the complexities of assessing impacts described in the previous paragraph.

2.3: 100 year was also selected as middle ground from IPCC FAR as values for 20, 100, and 500 years were given.

We modified the text as follows:

> The 100-year time horizon of the GWP (GWP100) is the horizon endorsed by the WMO and UNFCCCtime horizon most commonly used in many venues, for example in trading regimes such as under the Kyoto Protocol, perhaps in part because it was the middle value of the three time horizons (20, 100, and 500 years) analysed in the IPCC First Assessment Report.

2.8: Not sure why the word "therefore" is here. A description of why 100 year was selected does not in itself provide justification for why scientists are promoting 20 years. It is because 100 years does not adequately capture near-term impacts as it masks the importance of short-lived climate pollutants in the near-term. There needs to be a better transition from the 100 year discussion to the 20 year discussion.

We replaced "therefore" with "recently".

2.10: Papers such as Ocko et al. 2017 are not pushing for shortened time horizon, they are pushing for a two-valued GWP metric that includes BOTH 20 and 100 year time horizons. Very important distinction that needs to be clarified, as there are efforts (some livestock groups) that push for short time horizon only.

As noted above, we have attempted to present a more nuanced summary.

2.13: Part of the reason that other timescales are not suggested is because of the climate policy community's familiarity with 20 and 100 years. Just as they don't want to adopt a whole new metric, it is very plausible that they will reject a new time horizon. Since 20 and 100 years are adequate for near- and long-term, pushing for say 30 and 200 year time horizons may be counter-productive.

We have tried to clarify that we want more quantitative justifications of timescales in general (whether 20, 100, or 500, or anywhere in between).

2:20: There are more recent papers that need to be cited that look at the intersection of climate metrics and economics (Tol et al. 2012; Mallapragada and Mignone 2017). http://iopscience.iop.org/article/10.1088/1748-9326/7/4/044006/meta
http://iopscience.iop.org/article/10.1088/1748-9326/aa7397

See above for our language including some of these references and adding better context, and also our responses to other referees.

2.31: Why CO2 and CH4 only? Justification needed, such as represent the largest long-lived and short-lived climate pollutant contributor's to today's radiative forcing.

We have added the following sentence:

The paper focuses on CO2 and CH4 as the two most important historical anthropogenic contributors to current warming, but the methodology is applicable to emissions of other gases and sensitivity analyses consider N2O and some fluorinated gases.

3.5: Why is a pulse of 28.3 Mt of CH4 used, just bc of 10ppb? Why not today's annual emissions of methane from human activities (around 300-400 Mt)?

See our reply to William Collins for a description of the sensitivity analysis we performed in response to this comment, showing that sensitivity to the size of the pulse is small compared to other uncertainties.

3.12: What radiative efficiencies are used? Should specify this since you go into so much detail of other parameter values. I'm assuming radiative efficiencies are from IPCC AR5 but as you cite in your references, there are more recent calculations in Etminan et al. 2016.

The source of the radiative calculations is described here:

The perturbation of radiative forcing from additional GHG concentrations are based on the equations in Table 8.SM.1 from IPCC AR5. $CH_4$ forcing is adjusted by a factor of 1.65 to account for effects on tropospheric ozone and stratospheric water vapor, as is standard in GWP calculations. $N_2O$ forcing is adjusted by a factor of 0.928 to account for $N_2O$'s impacts on $CH_4$ concentrations, as is also standard in GWP calculations. Baseline radiative forcing is derived from the RCP scenario database.

As noted, Etminan has been cited as an example of updated information on radiative efficiency. Referee 3 also cited Modak et al. as showing the efficacy of methane forcing being lower than for $CO_2$ forcing.

See our reply to William Collins for a description of the sensitivity analysis we performed in response to this comment, showing that sensitivity to even a doubling of methane's radiative forcing would be very small compared to other uncertainties.

3:28: What damages are included by using this function?

This is meant as a simple approximation of all climate damages (sea level, health, ecosystems, etc.).

3:30: Please include citations for the first alternative.

We can cite the National Academies Social Cost of Carbon assessment here.

4.2: Suggest mentioning how these results fit in with scientific literature that has looked at these tradeoffs for decades.

We're not sure what tradeoffs the referee is referring to here: we discuss the Nordhaus GDP growth rates in the context of Gillingham et al. (now Christensen et al.).

**Reviewer 3 Response:**

5  I am not an expert in "Economics of climate change" and "discount rates" applied in estimating damages from climate change. Hence, I ask the editor to rely on the opinion of reviewers who are experts in assessing the economic damage from climate change. Here I am providing just a couple of minor comments.

We would like to thank the referee for their comments. For responses comment by comment, see below. All author replies are in red. There is also a summary of new sensitivity analyses that is included at the end of the reply to William Collins.

10  A recent paper (Modak, A., G. Bala, K. Caldeira, and L. Cao, 2018: Does shortwave absorption by Methane influence its effectiveness? Climate Dynamics, https://doi.org/10.1007/s00382-018-4102-x) shows that the efficacy of methane forcing is only 80% relative to $CO_2$ forcing. The lower efficacy affects the estimation of GTP and hence the damages. What is the implication of this result to the conclusion reached in your study? Discuss.

Modak et al. 2018 appears to be a similar paper, though opposite in direction, from Etminan et al. 2016 which was referred
15  to by Referee #2 and in the original paper. We have performed a sensitivity analysis wherein we double the radiative efficiency of methane. In this case, while damage ratios double, the GWP calculated with the updated radiative efficiency also doubles, such that the net effect on calculated timescales is less than $1/10^{th}$ of a percent (see sensitivity analysis in the reply to William Collins).

Modak et al. is slightly different than Etminan, as it occurs after radiative efficiency in the causal chain. However, it seems
20  likely that an updated GWP calculation for methane might take into account forcing efficacy, much the way it takes into account ozone and stratospheric water vapor perturbations.

In the abstract and in the 2nd paragraph of the Introduction section, it is stated that GWP assumes constant future concentrations. I believe this is true only for the baseline state. GWP is estimated for a case where the concentration of the gases decay with time. The integrated radiative forcing is calculated for the time evolving concentrations relative the
25  baseline.

The GWP assumes constant background concentrations, which is what the abstract and $2^{nd}$ paragraph refers to. An additional increment to the concentration at time zero from an emissions pulse is added, and this additional increment decays over time.

Fig. 1c and d: Is the unit for the damages and discounted damages correct? Should it be Billion $ per year? Same issue for Fig. S1

30  Correct. We will update the axis titles accordingly.

**Reviewer 4 Response:**

The manuscript makes a useful contribution to the literature by exploring explicitly how different time scales for GWP relate to GHG equivalence ratios based on damage costs and different discount rates. It is clearly written and highly readable. I have no fundamental concern with the technical approach and quantitative results, but I feel the manuscript needs to work a bit harder to develop its value proposition, discussion of results including sensitivity analysis, and finally the conclusions, before it is fit for publication.

We would like to thank the referee for their comments. For responses comment by comment, see below. All author replies are in red. There is also a summary of new sensitivity analyses that is included at the end of the reply to William Collins.

I'm comfortable with and largely endorse the comments already posted by Bill Collins and anonymous referee #2, and will try not to repeat the specific points they made.

My main concerns where I feel the manuscript needs to work harder are as follows:

1) value proposition: it is mainly in the SI that the authors acknowledge prior work that linked GHG equivalencies based on damage costs and discount rates to GWP. I believe this needs to be brought into the main paper up-front, and the authors need to do a better job explaining where their study adds value to those existing studies. For example, one could argue that their approach is simply a reverse reading of Boucher (2012). I don't think that accusation would be justified, but neither is it justifiable for the manuscript not to recognise the fact that a range of studies have already found that discount rates around 2-3% give the same GHG equivalence between CH4 and CO2 as GWP100. In this context, in the discussion, I would have liked to see a better explanation why their GWP100-equivalent discount rate of 3.3% is higher than that derived by both Boucher and Fuglestvedt et al.

We have brought much of the SI discussion into the main text and expanded upon it, particularly with the comparison to Boucher & Fuglestvedt. See our response to Collins. We have also determined that at least part of the difference between Boucher & Fuglestvedt and the current work is the assumption that damages are a percent of GDP rather than an absolute function of temperature, which increases the relative value of long-lived gases when assuming economic growth into the future.

2) discussion of results including sensitivity analysis: in my view, the authors should include an explicit simulation of results if climate-carbon cycle feedbacks following a pulse emission of CH4 are included. The IPCC AR5 and subsequent studies demonstrated that including this results in a significant increase in the GWP100. This is flagged (p7 of the manuscript) but appears not to have been included in the actual sensitivity analysis. It should be fairly easy to modify the radiative forcing calculations to simulate climate-carbon cycle feedbacks and it doesn't have to change the study design at all. There really is no good justification in my view not to include this, other than this is not how the GWP has been defined historically – but from a scientific consistency perspective, it makes no sense to include an effect for one gas (CO2) but not for the other. Including this in the sensitivity analysis (perhaps as a special case, since this is a binary choice rather than something that can be expressed via a pdf) would at least tell us how important this is when we are concerned about choosing GHG equivalencies based on damage functions and discount rates. I could even live with the authors running this only for a central estimate for all other parameters so we can get an order-of-magnitude sense.

See our description of sensitivity analyses at the end of the response to William Collins for a description of the sensitivity analysis we performed in response to this comment, showing that sensitivity to exclusion of the climate-carbon feedback from $CO_2$ had only a small effect. (as well as discussing why we chose that approach rather than inclusion of the climate-carbon feedback in the $CH_4$ effect). We agree that this was an important analysis to do. The effect of the exclusion was small, due to cancellation when the GWP and the damage ratio are both calculated using consistent assumptions about gas lifetimes and radiative efficiencies. This sensitivity analysis has been included in the manuscript.

Related to this, but more difficult to do (hence I would not insist that this is done quantitatively) is consideration of the rate of change as a source of damages. Again this could be parameterised and quantified, but there is a large degree of arbitrariness how much weight to place on rate of change vs amount of change. The manuscript would be much stronger though if it could demonstrate under what circumstances including the rate of change might affect the conclusions, or
5   whether the conclusions might be robust even if rate of change damages have been incorporated within reasonable bounds.

We have implemented a crude rate of change analysis within our framework (see discussion in the sensitivity analysis at the end of the reply to William Collins for details), and determined that under RCP6 inclusion of even extreme damage estimates due to rate of change have little effect on implicit timescales. However, under the RCP3PD scenario, the incorporation of rate of change has a larger effect, reducing the implicit timescale by almost half under the rate of change damage parameters
10   that may be more realistic in magnitude. Our approach is not sophisticated enough to become a major component of the paper, but we do have a brief discussion of this as a sensitivity analysis, along with reference to Bowerman (2013) and the finding that near-term reduction of SLCFs only reduces peak rate of change for stringent mitigation scenarios.

3) interpretation and conclusions: I would endorse some of the comments made by anonymous reviewer #2, that the authors are effectively beating up a strawman. Yes some people have argued that we should simply 'switch' to GWP20, but the more
15   intelligent arguments are all for considering the effect of multiple alternative time horizons to inform abatement decisions and policy choices. See e.g. the conclusions in Levasseur et al 2016 (doi: 10.1016/j.ecolind.2016.06.049) regarding the use of multiple time horizons and metrics in lifecycle assessment. The discussion and conclusions need to add quite a bit of nuance to reflect what those studies actually say, and hence the degree to which this manuscript challenges their conclusions or simply adds another dimension that can help choosing the right metric for the right purpose.

20   We have added more nuance to our description (see responses to other referees), as well as additional context to our results.

There are two additional points that the discussion and conclusion needs to address:

(a) one is that a key context in which GHG metrics are used are in emission trading schemes, and to help governments evaluate policy choices that directly affect near- term commercial decisions, i.e. policy that would "alter the use of capital in the private sector". So there are very different contexts in which GHG metrics are actually used in climate policy and where
25   different discount rates are commonly applied, and the paper would be stronger and more relevant if it recognised and addressed these explicitly.

We recognize that there is common justification (e.g., OMB Circular A-4) for the use of one discount rate for social problems and another for policies that "alter the use of capital in the private sector". We would argue, however, that comparing the relative impacts of $CO_2$ v. $CH_4$ should always been considered a social problem in this context, regardless of
30   whether it is being used in decisions regarding capital or by governments within emission trading programs. This is a somewhat tricky topic, as it can pose consistency challenges that go well beyond the implications of this paper. For example, a decisionmaker deciding between investing in a CNG vehicle and a gasoline vehicle might want to look at vehicle costs, maintenance, and fuel prices under a high discount rate to reflect the opportunity cost of that investment, but, because the benefits of the GHG abatement are not received by the decisionmaker but by society as a whole, it could be argued that the
35   latter issue should be considered with a societal discount rate. It is unclear how to bring those two monetization streams into a single analysis given the difference in discount rates.

(b) The second is a recognition that IAMs used to design cost-minimising emission pathways often use a discount rate of 5%. Given that a(nother) key use of GHG metrics is to help IAMs make trade-offs between different gases with different mitigation costs, this should enter into the discussion in this paper. I don't think this materially changes the conclusions since
40   we know that different GHG metrics don't have a massive effect on total mitigation costs (although there is a systematic effect especially when moving towards GWP20), but the issue is not trivial especially for countries or sectors with non-negligible non-CO2/SLCF sources. Some discussion on this is needed.

We plan to add more discussion regarding some of the IAM-based tradeoff work such as van den Berg et al. (2015), Reisinger et al. (2013), and Smith et al. (2013). The common use of a 5% discount rate is now briefly mentioned in that context.

I believe that all the above points (with the exception of quantifying the effect of including climate-carbon cycle feedbacks for CH4) can be addressed by a careful revision of the text itself. The manuscript needs to avoid what currently appears as the too- simplistic conclusion that "actually, GWP100 is largely ok, let's move on" (which is how I read P8L22). The fundamental finding from virtually all metrics papers is that the right metric depends on the application, and hence it is rather jarring to read a conclusion that continued use of GWP100 is 'reasonable' without any caveat.

We have made a number of changes to the conclusion to better explain our position. While we do believe that this analysis is strong support for a 100 year GWP in many contexts, it is important to define the key assumptions involved in that finding, including the fact that there are some uses of metrics for which the structure of this analysis may not be appropriate.

I am not repeating the above points in my specific comments below and would be happy for the authors to decide how they can best address them.

Specific comments:

P1L22: insert 'emission' after gases – we're talking about emission metrics here

Done.

P2L3: 'endorsed' is too strong in my view for the UNFCCC – 'used' is more factual, I cannot recall an explicit endorsement in the sense that the UNFCCC would have explained and justified its choice.

Discussion updated to:

> The 100-year time horizon of the GWP (GWP100) is the time horizon most commonly used in many venues, for example in trading regimes such as under the Kyoto Protocol, perhaps in part because it was the middle value of the three time horizons (20, 100, and 500 years) analysed in the IPCC First Assessment Report.

P2L11: I believe the correct term for GTP is Global Temperature CHANGE Potential

In our defense, AR4 and other sources also refer to the GTP as the Global Temperature Potential: e.g., "2.10.4.2 The Global Temperature Potential"). But "change" seems to be more standard, including in AR5, and so the manuscript has been updated to use the correct terminology.

P2L12: the reason why GTP downplays SLCFs is not primarily that it is temperature based but that it is a point metric. iGTP is very similar to GWP.

We are updating language to better differentiate integrated versus endpoint metrics. We note, however, that Brazil and New Zealand specifically suggest the GTP, not the iGTP, and justify it based on the temperature argument.

P2L22-27: editorial only: I prefer if introductions don't include the conclusions but rather focus on making the point of why the conclusions are worth having.

We have edited accordingly.

P3L15: shouldn't the N2O effect on CH4 forcing depend on the RCP pathway? Perhaps this was done but this isn't clear to me from the text.

The particular adjustment for $N_2O$ cited here is based on 8.SM.11.3.3, which estimates that emissions of 100 molecules of $N_2O$ will lead to a destruction of 36 molecules of CH4. For this effect, the RCP pathway does not matter.

This is in contrast to the overlap between $N_2O$ and $CH_4$ radiative absorption bands, which does depend on the RCP pathway, and is captured in the radiative forcing equations used for this paper (and for the GWP).

P4L9: 'future years are cooler than present': helpful if you could indicate what years we are talking about (presumably you mean after 2200 or thereabouts, depending on the reference period/warming – meaning that much of those will be heavily discounted anyway).

Under RCP3PD, with climate sensitivity of 3.92, and a forcing imbalance of 0.84, temperatures drop below the starting temperature only 458 years into the analysis. Therefore, in this case, when the damage function is relying on temperature change since present (rather than since 1950-1980 or earlier), these years would need to be set to zero. It is possible that under different climate sensitivities or forcing imbalances, this could occur somewhat earlier, but we believe that the referee's intuition that this effect will be negligible due to discounting is correct. We have included a sentence noting that this scenario occurs in "fewer than 1 out of 1000 of the total years considered across all sensitivities, and generally only for years near the end of the analysis."

P4L21: here and later, please clarify where you truncate your damage calculations (when I read this sentence, I thought you truncate at 2300, but later (P5L14) it seems you truncate at 2500). You note below that this may matter for very low discount rates. Can you quantify/illustrate this?

We have clarified that the graphs are to 2300, but the calculations do go to 2500. We also include text regarding the size of the effect as follows:

> Even at a 2% discount rate, 95% of the $CO_2$ damages come in the first 287 years. At discount rates lower than 2%, however, truncation effects can account for errors in damage ratio estimates of greater than a percent, indicating that longer calculation timeframes may be necessary to capture the full effect of the emissions pulse.

P6L4: I think the entire sensitivity discussion should note that projecting damages multiple centuries into the future is increasingly fraught with difficulties. The AR5 chose not to evaluate GWP500 because the authors felt that (deep) uncertainties were simply too large – but here you evaluate damages from temperature responses from forcing 500 years into the future? At least a comment on this is needed here – the discussion of what percentage of total damages occurs up to a given year for CH4 and CO2 is useful in this context and could be linked to this point about uncertainty.

See the response above regarding the low percentage of damages that occur after the first 290 years even under a 2% discount rate. We have also included a discussion of this issue:

> Myhre et al. (2013) justified exclusion of the 500 year GWP based on the large uncertainties and ambiguities involved with far future projections. This analysis extends through 2500, and therefore might be subject to some of those same uncertainties. Therefore, the effect of two shorter time periods were investigated. When truncating the analysis after 150 years, the GWP100 was still found to be consistent with a discount rate of 3.3%, with the upper interquartile bound also remaining constant at 4.1%, though the lower end of the interquartile range decreased

modestly to 2.4%. When the analysis was truncated at 100 years into the future, the implicit discount rates dropped more substantially, to 2.6% (interquartile range of 1.5% to 3.5%). Truncating the analysis will naturally make CH4 mitigation appear more favourable relative to CO2, but even discount rates as small as 3% are sufficient to make effects more than 150 years into the future inconsequential to the results.

P8L2: I feel the statement "We note no metric designed to tradeoff emission pulses is consistent with stabilization" is too strong. Of course, no metric in itself delivers stabilisation, but almost any metric can be used wisely enough to help countries achieve stabilisation.

We have modified our discussion of this issue. We particularly wanted to highlight the challenge of using, for example, a GTPX metric to achieve a cost-optimal approach to achieving a temperature target in a given year, but which when used in that way would lead to increasing challenges of maintaining temperatures at that target in future years (in part because the short-term GTP would lead to more SLCF abatement relative to $CO_2$ abatement than would be indicated by, for example, a GWP100). This is in addition to the fact that any SCLF/$CO_2$ trading using any pulse-based metric post-stabilization will lead to moving away from stabilization. The development of the GWP* in Allen et al. 2016 is one possible response to this challenge.

**List of relevant changes:**

Three substantial changes were made to the original manuscript:

1) A more complete & nuanced description of the existing literature in the introduction.
2) A number of additional sensitivity analyses exploring issues of discouting, climate-carbon feedbacks, pulse sizes, radiative efficiency questions, Ramsey discounting, and rate-of-change damage functions.
3) The conclusion has been rewritten in order to reflect the additional nuances, better characterization of existing literature, and new sensitivity analyses.

In addition, a number of minor edits have been made throughout. A graph of GDPs has been added to the SI (and the portions of the SI that were brought into the main text have been deleted).

**Marked-up manuscript (see below):**

[revised manuscript text omitted]